# Selection via Proxy: Efficient Data Selection for Deep Learning

**Cody Coleman,**[*] **Christopher Yeh, Stephen Mussmann, Baharan Mirzasoleiman,**
**Peter Bailis, Percy Liang, Jure Leskovec, Matei Zaharia**
Stanford University

## Abstract

Data selection methods, such as active learning and core-set selection, are useful tools for machine learning on large datasets. However, they can be prohibitively expensive to apply in deep learning because they depend on feature representations that need to be learned. In this work, we show that we can greatly improve the computational efficiency by using a small proxy model to perform data selection (e.g., selecting data points to label for active learning). By removing hidden layers from the target model, using smaller architectures, and training for fewer epochs, we create proxies that are an order of magnitude faster to train. Although these small proxy models have higher error rates, we find that they empirically provide useful signals for data selection. We evaluate this "selection via proxy" (SVP) approach on several data selection tasks across five datasets: CIFAR10, CIFAR100, ImageNet, Amazon Review Polarity, and Amazon Review Full. For active learning, applying SVP can give an order of magnitude improvement in data selection runtime (i.e., the time it takes to repeatedly train and select points) without significantly increasing the final error (often within 0.1%). For core-set selection on CIFAR10, proxies that are over $10\times$ faster to train than their larger, more accurate targets can remove up to 50% of the data without harming the final accuracy of the target, leading to a $1.6\times$ end-to-end training time improvement.

## 1 Introduction

Data selection methods, such as active learning and core-set selection, improve the *data efficiency* of machine learning by identifying the most informative training examples. To quantify informativeness, these methods depend on semantically meaningful features or a trained model to calculate uncertainty. Concretely, active learning selects points to label from a large pool of unlabeled data by repeatedly training a model on a small pool of labeled data and selecting additional examples to label based on the model's uncertainty (e.g., the entropy of predicted class probabilities) or other heuristics (Lewis & Gale, 1994; Rosenberg et al., 2005; Settles, 2011; 2012). Conversely, core-set selection techniques start with a large labeled or unlabeled dataset and aim to find a small subset that accurately approximates the full dataset by selecting representative examples (Har-Peled & Kushal, 2007; Tsang et al., 2005; Huggins et al., 2016; Campbell & Broderick, 2017; 2018; Sener & Savarese, 2018).

Unfortunately, classical data selection methods are often prohibitively expensive to apply in deep learning (Shen et al., 2017; Sener & Savarese, 2018; Kirsch et al., 2019). Deep learning models learn complex internal semantic representations (hidden layers) from raw inputs (e.g., pixels or characters) that enable them to achieve state-of-the-art performance but result in substantial training times. Many core-set selection and active learning techniques require some feature representation *before* they can accurately identify informative points either to take diversity into account or as part of a trained model to quantify uncertainty. As a result, new deep active learning methods request labels in large batches to avoid retraining the model too many times (Shen et al., 2017; Sener & Savarese, 2018; Kirsch et al., 2019). However, batch active learning still requires training a full deep model for every batch, which is costly for large models (He et al., 2016b; Jozefowicz et al., 2016; Vaswani et al., 2017). Similarly, core-set selection applications mitigate the training time of deep learning models by using

---

[*]Correspondence: cody@cs.stanford.edu

bespoke combinations of hand-engineered features and simple models (e.g., hidden Markov models) pretrained on auxiliary tasks (Wei et al., 2013; 2014; Tschiatschek et al., 2014; Ni et al., 2015).

In this paper, we propose *selection via proxy (SVP)* as a way to make existing data selection methods more computationally efficient for deep learning. SVP uses the feature representation from a separate, less computationally intensive proxy model in place of the representation from the much larger and more accurate target model we aim to train. SVP builds on the idea of heterogeneous uncertainty sampling from Lewis & Catlett (1994), which showed that an inexpensive classifier (e.g., naïve Bayes) can select points to label for a much more computationally expensive classifier (e.g., decision tree). In our work, we show that small deep learning models can similarly serve as an inexpensive proxy for data selection in deep learning, significantly accelerating both active learning and core-set selection across a range of datasets and selection methods. To create these cheap proxy models, we can scale down deep learning models by removing layers, using smaller model architectures, and training them for fewer epochs. While these scaled-down models achieve significantly lower accuracy than larger models, we surprisingly find that they still provide useful representations to rank and select points. Specifically, we observe high Spearman's and Pearson's correlations between the rankings from small proxy models and the larger, more accurate target models on metrics including uncertainty (Settles, 2012), forgetting events (Toneva et al., 2019), and submodular algorithms such as greedy k-centers (Wolf, 2011). Because these proxy models are quick to train (often $10\times$ faster), we can identify which points to select nearly as well as the larger target model but significantly faster.

We empirically evaluated SVP for active learning and core-set selection on five datasets: CIFAR10, CIFAR100 (Krizhevsky & Hinton, 2009), ImageNet (Russakovsky et al., 2015), Amazon Review Polarity, and Amazon Review Full (Zhang et al., 2015). For active learning, we considered both least confidence uncertainty sampling (Settles, 2012; Shen et al., 2017; Gal et al., 2017) and the greedy k-centers approach from Sener & Savarese (2018) with a variety of proxies. Across all datasets, we found that SVP matches the accuracy of the traditional approach of using the same large model for both selecting points and the final prediction task. Depending on the proxy, SVP yielded up to a $7\times$ speed-up on CIFAR10 and CIFAR100, $41.9\times$ speed-up on Amazon Review Polarity and Full, and $2.9\times$ speed-up on ImageNet in data selection runtime (i.e., the time it takes to repeatedly train and select points). For core-set selection, we tried three methods to identify a subset of points: max entropy uncertainty sampling (Settles, 2012), greedy k-centers as a submodular approach (Wolf, 2011), and the recent approach of forgetting events (Toneva et al., 2019). For each method, we found that smaller proxy models have high Spearman's rank-order correlations with models that are $10\times$ larger and performed as well as these large models at identifying subsets of points to train on that yield high test accuracy. On CIFAR10, SVP applied to forgetting events removed 50% of the data without impacting the accuracy of ResNet164 with pre-activation (He et al., 2016b), using a $10\times$ faster model than ResNet164 to make the selection. This substitution yielded an end-to-end training time improvement of about $1.6\times$ for ResNet164 (including the time to train and use the proxy). Taken together, these results demonstrate that SVP is a promising, yet simple approach to make data selection methods computationally feasible for deep learning. While we focus on active learning and core-set selection, SVP is widely applicable to methods that depend on learned representations.

## 2 METHODS

In this section, we describe SVP and show how it can be incorporated into active learning and core-set selection. Figure 1 shows an overview of SVP: in active learning, we retrain a proxy model $A_k^P$ in place of the target model $A_k^T$ after each batch is selected, and in core-set selection, we train the proxy $A_{[n]}^P$ rather than the target $A_{[n]}^T$ over all the data to learn a feature representation and select points.

### 2.1 ACTIVE LEARNING

Pool-based active learning starts with a large pool of unlabeled data $U = \{\mathbf{x}_i\}_{i \in [n]}$ where $[n] = \{1, \ldots, n\}$. Each example is from the space $\mathcal{X}$ with an unknown label from the label space $\mathcal{Y}$ and is sampled *i.i.d.* over the space $\mathcal{Z} = \mathcal{X} \times \mathcal{Y}$ as $(\mathbf{x}_i, y_i) \sim p_{\mathcal{Z}}$. Initially, methods label a small pool of points $s^0 = \{s_j^0 \in [n]\}_{j \in [m]}$ chosen uniformly at random. Given $U$, a loss function $\ell$, and the labels $\{y_{s_j^0}\}_{j \in [m]}$ for the initial random subset, the goal of active learning is to select up to a budget of $b$ points $s = s^0 \cup \{s_j \in [n] \setminus s^0\}_{j \in [b-m]}$ to label that produces a model $A_s$ with low error.

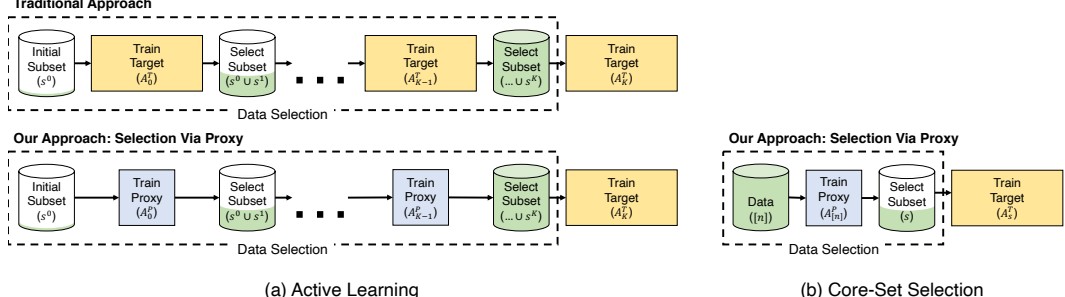

(a) Active Learning                    (b) Core-Set Selection

Figure 1: **SVP applied to active learning** (left) **and core-set selection** (right). In active learning, we followed the same iterative procedure of training and selecting points to label as traditional approaches but replaced the target model with a cheaper-to-compute proxy model. For core-set selection, we learned a feature representation over the data using a proxy model and used it to select points to train a larger, more accurate model. In both cases, we found the proxy and target model have high rank-order correlation, leading to similar selections and downstream results.

**Baseline.** In this paper, we apply SVP to least confidence uncertainty sampling (Settles, 2012; Shen et al., 2017; Gal et al., 2017) and the recent greedy k-centers approach from Sener & Savarese (2018). Like recent work for deep active learning (Shen et al., 2017; Sener & Savarese, 2018; Kirsch et al., 2019), we consider a batch setting with $K$ rounds where we select $\frac{b}{K}$ points in every round. Following Gal et al. (2017); Sener & Savarese (2018); Kirsch et al. (2019), we reinitialize the target model and retrain on all of the labeled data from the previous $k$ rounds to avoid any correlation between selections (Frankle & Carbin, 2018; Kirsch et al., 2019). We denote this trained model as $A_{s^0 \cup \ldots \cup s^k}^T$ or just $A_k^T$ for simplicity. Then using $A_k^T$, we either calculate the model's confidence as:

$$f_{\text{confidence}}(\mathbf{x}; A_k^T) = 1 - \max_{\hat{y}} P(\hat{y}|\mathbf{x}; A_k^T)$$

and select the examples with the lowest confidence or extract a feature representation from the model's final hidden layer and compute the distance between examples (i.e., $\Delta(\mathbf{x}_i, \mathbf{x}_j; A_k^T)$) to select points according to the greedy k-centers method from Wolf (2011); Sener & Savarese (2018) (Algorithm 1). The same model is trained on the final $b$ labeled points to yield the final model, $A_K^T$, which is then tested on a held-out set to evaluate error and quantify the quality of the selected data.

Although other selection approaches exist, least confidence uncertainty sampling and greedy k-centers cover the spectrum of uncertainty-based and representativeness-based approaches for deep active learning. Other uncertainty metrics such as entropy or margin were highly correlated with confidence when using the same trained model (i.e., above a 0.96 Spearman's correlation in our experiments on CIFAR). Query-by-committee (Seung et al., 1992) can be prohibitively expensive in deep learning, where training a single model is already costly. BALD (Houlsby et al., 2011) has seen success in deep learning (Gal et al., 2017; Shen et al., 2017) but is restricted to Bayesian neural networks or networks with dropout (Srivastava et al., 2014) as an approximation (Gal & Ghahramani, 2016).

---

**Algorithm 1** GREEDY K-CENTERS
(WOLF, 2011; SENER & SAVARESE, 2018)

**Input:** data $\mathbf{x}_i$, existing pool $s^0$, trained model $A_0^T$, and a budget $b$
1: Initialize $s = s^0$
2: **repeat**
3:    $u = \arg\max_{i \in [n] \setminus s} \min_{j \in s} \Delta\left(\mathbf{x}_i, \mathbf{x}_j; A_0^T\right)$
4:    $s = s \cup \{u\}$
5: **until** $|s| = b + |s^0|$
6: **return** $s \setminus s^0$

---

**Algorithm 2** FORGETTING EVENTS
(TONEVA ET AL., 2019)

1: Initialize $\text{prev\_acc}_i = 0, i \in [n]$
2: Initialize $\text{forgetting\_events}_i = 0, i \in [n]$
3: **while** training is not done **do**
4:    Sample mini-batch $B$ from $L$
5:    **for** example $i \in B$ **do**
6:      Compute $\text{acc}_i$
7:      **if** $\text{prev\_acc}_i > \text{acc}_i$ **then**
8:        $\text{forgetting\_events}_i \mathrel{+}= 1$
9:      $\text{prev\_acc}_i = \text{acc}_i$
10:    Gradient update classifier on $B$
11: **return** forgetting\_events

## 2.2 CORE-SET SELECTION

Core-set selection can be broadly defined as techniques that find a subset of data points that maintain a similar level of quality (e.g., generalization error of a trained model or minimum enclosing ball) as the full dataset. Specifically, we start with a labeled dataset $L = \{\mathbf{x}_i, y_i\}_{i \in [n]}$ sampled *i.i.d.* from $\mathcal{Z}$ with $p_{\mathcal{Z}}$ and want to find a subset of $m \leq n$ points $s = \{s_j \in [n]\}_{j \in [m]}$ that achieves comparable quality to the full dataset: $\min_{s:|s|=m} E_{\mathbf{x},y \sim p_{\mathcal{Z}}} [\ell(\mathbf{x}, y; A_s)] - E_{\mathbf{x},y \sim p_{\mathcal{Z}}} [\ell(\mathbf{x}, y; A_{[n]})]$

**Baseline.** To find $s$ for a given budget $m$, we implement three core-set selection techniques: greedy k-centers (Wolf, 2011; Sener & Savarese, 2018), forgetting events (Toneva et al., 2019), and max entropy uncertainty sampling (Lewis & Gale, 1994; Settles, 2012). Greedy k-centers is described above and in Algorithm 1. Forgetting events are defined as the number of times an example is incorrectly classified after having been correctly classified earlier during training a model, as described in Algorithm 2. To select points, we follow the same procedure as Toneva et al. (2019): we keep the points with the $m$ highest number of forgetting events. Points that are never correctly classified are treated as having an infinite number of forgetting events. Similarly, we rank examples based on the entropy from a trained target $A_{[n]}^T$ as:

$$f_{\text{entropy}}(\mathbf{x}; A_{[n]}^T) = -\sum_{\hat{y}} P(\hat{y}|\mathbf{x}; A_{[n]}^T) \log P(\hat{y}|\mathbf{x}; A_{[n]}^T)$$

and keep the $m$ examples with the highest entropy. To evaluate core-set quality, we compare the performance of training the large target model on the selected subset $A_s^T$ to training the target model on the entire dataset $A_{[n]}^T$ by measuring error on a held-out test set.

## 2.3 APPLYING SELECTION VIA PROXY

In general, SVP can be applied by replacing the models used to compute data selection metrics such as uncertainty with proxy models. In this paper, we applied SVP to the active learning and core-set selection methods described in Sections 2.1 and 2.2 as follows:

- For active learning, we replaced the model trained at each batch ($A_k^T$) with a proxy ($A_k^P$), but then trained the same final model $A_K^T$ once the budget $b$ was reached.
- For core-set selection, we used a proxy $A_{[n]}^P$ instead of $A_{[n]}^T$ to compute metrics and select $s$.

We explored two main methods to create our proxy models:

**Scaling down.** For deep models with many layers, reducing the dimension or the number of hidden layers is an easy way to trade-off accuracy to reduce training time. For example, deep ResNet models come in a variety of depths (He et al., 2016b;a) and widths (Zagoruyko & Komodakis, 2016) that represent many points on the accuracy and training time curve. As shown in Figure 4a in the Appendix, a ResNet20 model achieves a top-1 error of 7.6% on CIFAR10 in 26 minutes, while a larger ResNet164 model takes 4 hours and reduces error by 2.5%. Similar results have been shown for many other tasks, including neural machine translation (Vaswani et al., 2017), text classification (Conneau et al., 2016), and recommendation (He et al., 2017). Looking across architectures gives even more options to reduce computational complexity. We exploit the limitless model architectures in deep learning to trade-off between accuracy and complexity to scale down to a proxy that can be trained quickly but still provides a good approximation of the target's decision boundary.

**Training for fewer epochs.** Similarly, a significant amount of training is spent on a relatively small reduction in error. While training ResNet20, almost half of the training time (i.e., 12 minutes out of 26 minutes) is spent on a 1.4% improvement in test error, as shown in Figure 4a in the Appendix. Based on this observation, we also explored training proxy models for a smaller number of epochs to get good approximations of the decision boundary of the target model even faster.

## 3 RESULTS

We applied SVP to data selection methods from active learning and core-set selection on five datasets. After a brief description of the datasets and models in Section 3.1, Section 3.2 evaluates SVP's impact on active learning and shows that across labeling budgets SVP achieved similar or higher accuracy

and up to a $41.9\times$ improvement in data selection runtime (i.e., the time it takes to repeatedly train and select points). Next, we applied SVP to the core-set selection problem (Section 3.3). For all selection methods, the target model performed nearly as well as or better with SVP than the oracle that trained the target model on all of the data before selecting examples. On CIFAR10, a small proxy model trained for 50 epochs instead of 181 epochs took only 7 minutes compared to the 4 hours for training the target model for all 181 epochs, making SVP feasible for end-to-end training time speed-ups. Finally, Section 3.4 illustrates why proxy models performed so well by evaluating how varying models and methods rank examples.

## 3.1 Experimental Setup

**Datasets.** We focused on classification as a well-studied task in the active learning literature (see Section A.1 for more detail). Our experiments included three image classification datasets: CIFAR10, CIFAR100 (Krizhevsky & Hinton, 2009), and ImageNet (Russakovsky et al., 2015); and two text classification datasets: Amazon Review Polarity and Full (Zhang & LeCun, 2015; Zhang et al., 2015). CIFAR10 is a coarse-grained classification task over 10 classes, and CIFAR100 is a fine-grained task with 100 classes. Both datasets contain 50,000 images for training and 10,000 images for testing. ImageNet has 1.28 million training images and 50,000 validation images that belong to 1 of 1,000 classes. Amazon Review Polarity has 3.6 million reviews split evenly between positive and negative ratings with an additional 400,000 reviews for testing. Amazon Review Full has 3 million reviews split evenly between the 5 stars with an additional 650,000 reviews for testing.

**Models.** For CIFAR10 and CIFAR100, we used ResNet164 with pre-activation from He et al. (2016b) as our large target model. The smaller, proxy models are also ResNet architectures with pre-activation, but they use pairs of $3 \times 3$ convolutional layers as their residual unit rather than bottlenecks. For ImageNet, we used the original ResNet architecture from He et al. (2016a) implemented in PyTorch [1] (Paszke et al., 2017) with ResNet50 as the target and ResNet18 as the proxy. For Amazon Review Polarity and Amazon Review Full, we used VDCNN (Conneau et al., 2017) and fastText (Joulin et al., 2016) with VDCNN29 as the target and fastText and VDCNN9 as proxies. In general, we followed the same training procedure as the original papers (more details in Section A.2).

## 3.2 Active Learning

We explored the impact of SVP on two active learning techniques: least confidence uncertainty sampling and the greedy k-centers approach from Sener & Savarese (2018). Starting with an initial random subset of 2% of the data, we selected 8% of the remaining unlabeled data for the first round and 10% for subsequent rounds until the labeled data reached the budget $b$ and retrained the models from scratch between rounds as described in Section 2.1. Across datasets, SVP sped up data selection without significantly impacting the final predictive performance of the target.

**CIFAR10 and CIFAR100.** For least confidence uncertainty sampling and greedy k-centers, SVP sped-up data selection by up to $7\times$ and $3.8\times$ respectively without impacting data efficiency (see Tables 1 and 3) despite the proxy achieving substantially higher top-1 error than the target ResNet164 model (see Figure 6 in the Appendix). The speed-ups for least confidence were a direct reflection of the difference in training time between the proxy in the target models. As shown in Figures 4 and 5 in the Appendix, ResNet20 was about $8\times$ faster to train than ResNet164, taking 30 minutes to train rather than 4 hours. Larger budgets required more rounds of selection and, in turn, more training, which led to larger speed-ups as training became a more significant fraction of the total time. Training for fewer epochs provided a significant error reduction compared to random sampling but was not as good as training for the full schedule (see Table 4 in the Appendix). For greedy k-centers, the speed-ups increased more slowly because executing the selection algorithm added more overhead.

**ImageNet.** For least confidence uncertainty sampling, SVP sped-up data selection by up to $1.6\times$ (Table 1) despite ResNet18's higher error compared to ResNet50 (Figure 6g in the Appendix). Training for fewer epochs increased the speed-up to $2.9\times$ without increasing the error rate of ResNet50 (Table 4). Greedy k-centers was too slow on ImageNet due to the quadratic complexity of Algorithm 1.

---

[1] https://pytorch.org/docs/stable/torchvision/models.html

Table 1: **SVP performance on active learning.** Average ($\pm$ 1 std.) data selection speed-ups from 3 runs of active learning using least confidence uncertainty sampling with varying proxies and labeling budgets on four datasets. **Bold** speed-ups indicate settings that either achieve lower error or are within 1 std. of the mean top-1 error for the baseline approach of using the same model for selection and the final predictions. Across datasets, SVP sped up selection without significantly increasing the error of the final target. Additional results and details are in Table 3.

| | | Data Selection Speed-up | | | | |
|---|---|---|---|---|---|---|
| | **Budget** ($b/n$) | 10% | 20% | 30% | 40% | 50% |
| **Dataset** | **Selection Model** | | | | | |
| CIFAR10 | ResNet164 (Baseline) | **1.0**$\times$ | **1.0**$\times$ | **1.0**$\times$ | **1.0**$\times$ | **1.0**$\times$ |
| | ResNet110 | **1.8**$\times$ | **1.9**$\times$ | **1.9**$\times$ | **1.8**$\times$ | **1.8**$\times$ |
| | ResNet56 | **2.6**$\times$ | **2.9**$\times$ | **3.0**$\times$ | **3.1**$\times$ | 3.1$\times$ |
| | ResNet20 | **3.8**$\times$ | **5.8**$\times$ | **6.7**$\times$ | **7.0**$\times$ | 7.2$\times$ |
| CIFAR100 | ResNet164 (Baseline) | **1.0**$\times$ | **1.0**$\times$ | **1.0**$\times$ | **1.0**$\times$ | **1.0**$\times$ |
| | ResNet110 | **1.5**$\times$ | **1.6**$\times$ | **1.6**$\times$ | **1.6**$\times$ | 1.6$\times$ |
| | ResNet56 | **2.4**$\times$ | **2.7**$\times$ | **3.0**$\times$ | **2.9**$\times$ | **3.1**$\times$ |
| | ResNet20 | 4.0$\times$ | **5.8**$\times$ | **6.6**$\times$ | **7.0**$\times$ | **7.2**$\times$ |
| ImageNet | ResNet50 (Baseline) | **1.0**$\times$ | **1.0**$\times$ | **1.0**$\times$ | **1.0**$\times$ | **1.0**$\times$ |
| | ResNet18 | **1.2**$\times$ | **1.3**$\times$ | **1.4**$\times$ | **1.5**$\times$ | **1.6**$\times$ |
| Amazon | VDCNN29 (Baseline) | **1.0**$\times$ | **1.0**$\times$ | **1.0**$\times$ | **1.0**$\times$ | **1.0**$\times$ |
| Review | VDCNN9 | **1.9**$\times$ | 1.8$\times$ | **1.8**$\times$ | **1.8**$\times$ | 1.8$\times$ |
| Polarity | fastText | 10.6$\times$ | 20.6$\times$ | 32.2$\times$ | **41.9**$\times$ | 51.3$\times$ |

**Amazon Review Polarity and Amazon Review Full.** On Amazon Review Polarity, SVP with a fastText proxy for VDCNN29 led to up to a relative error reduction of 14% over random sampling for large budgets (Table 3), while being up to $41.9\times$ faster at data selection than the baseline approach (Table 1). Despite fastText's architectural simplicity compared to VDCNN29 and higher error (Figure 6e), the calculated confidences signaled which examples would be the most informative. For all budgets, VDCNN9 was within 0.1% top-1 error of VDCNN29, giving a consistent $1.8\times$ speed-up. On Amazon Review Full, neither the baseline least confidence uncertainty sampling approach nor the application of SVP outperformed random sampling (see Table 3 in the Appendix), so the data selection speed-ups were uninteresting even though they were similar to Amazon Review Polarity. For both datasets, greedy k-centers was too slow as mentioned above in the ImageNet experiments.

## 3.3 CORE-SET SELECTION

**CIFAR10 and CIFAR100.** For all methods on both CIFAR10 and CIFAR100, SVP proxy models performed as well as or better than an oracle where ResNet164 itself is used as the core-set selection model, as shown in Figure 2 (and Figure 7 in the Appendix). Using forgetting events on CIFAR10, SVP with ResNet20 as the proxy removed 50% of the data without a significant increase in error from ResNet164. The entire process of training ResNet20 on all the data, selecting which examples to keep, and training ResNet164 on the subset only took 2 hours and 20 minutes (see Table 6 in the Appendix), which was a $1.6\times$ speed-up compared to training ResNet164 over all of the data. If we stopped training ResNet56 early and removed 50% of the data based on forgetting events from the first 50 epochs, SVP achieved an end-to-end training time speed-up of $1.8\times$ with only a slightly higher top-1 error from ResNet164 (5.4% vs. 5.1%) as shown in Table 7 in the Appendix. In general, training the proxy for fewer epochs also maintained the accuracy of the target model on CIFAR10 because the ranking quickly converged (Figure 11a and 12a in the Appendix). On CIFAR100, partial training did not work as well for proxies at large subset sizes because the ranking took longer to stabilize and were less correlated (Figure 11b and Figure 12b in the Appendix). On small 30% subsets with forgetting events, partial training improved accuracy on CIFAR100.

**ImageNet.** Neither the baseline approach nor SVP was able to remove a significant percentage of the data without increasing the final error of ResNet50, as shown in Table 5 in the Appendix. However, the selected subsets from both ResNet18 and ResNet50 outperformed random sampling with up to a 1% drop in top-1 error using forgetting events. Note, due to the quadratic computational complexity of Algorithm 1, we were unable to run greedy k-centers in a reasonable amount of time.

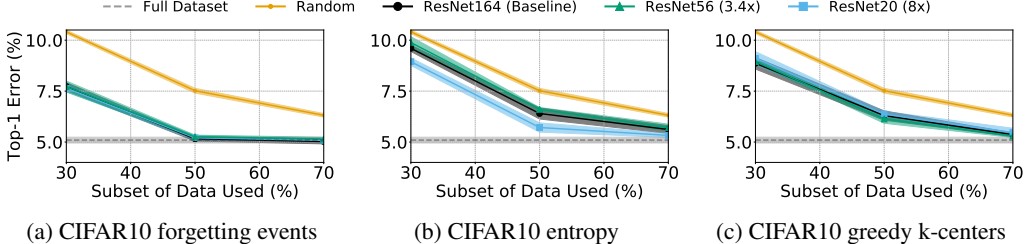

(a) CIFAR10 forgetting events    (b) CIFAR10 entropy    (c) CIFAR10 greedy k-centers

Figure 2: **SVP performance on core-set selection.** Average ($\pm$ 1 std.) top-1 error of ResNet164 over 5 runs of core-set selection with different selection methods, proxies, and subset sizes on CIFAR10. We found subsets using forgetting events (left), entropy (middle), and greedy k-centers (right) from a proxy model trained over the entire dataset. Across datasets and selection methods, SVP performed as well as an oracle baseline but significantly faster (speed-ups in parentheses).

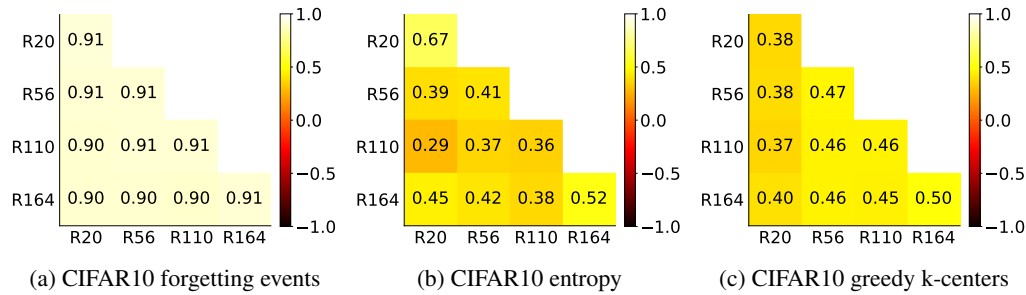

(a) CIFAR10 forgetting events    (b) CIFAR10 entropy    (c) CIFAR10 greedy k-centers

Figure 3: **Comparing selection across model sizes and methods on CIFAR10.** Average Spearman's correlation between different runs of ResNet (R) models and at varying depths. We computed rankings based on forgetting events (left), entropy (middle), and greedy k-centers (right). We saw a similarly high correlation across model architectures (*off-diagonal*) as between different runs of the same architecture (*on-diagonal*), suggesting that small models are good proxies for data selection.

**Amazon Review Polarity and Amazon Review Full**. On Amazon Review Polarity, we were able to remove 20% of the dataset with only a 0.1% increase in VDCNN29's top-1 error using fastText as the proxy (see Table 5). In comparison to VDCNN29, which took 16 hours and 40 minutes to train over the entire dataset on a Titan V GPU, fastText was two orders of magnitude faster, taking less than 10 minutes on a CPU to train over the same data and compute output probabilities. This difference allowed us to train VDCNN29 to nearly the same error in 13 and a half hours. However, on Amazon Review Full, both the baseline approach and SVP failed to outperform random sampling. Similar to ImageNet, we were unable to run greedy k-centers in a reasonable amount of time, and additionally, Facebook's fastText implementation [2] did not allow us to compute forgetting events.

## 3.4    RANKING CORRELATION BETWEEN MODELS

**Models with fewer layers.** Figure 3 (and Figure 9 in the appendix) shows the Spearman's rank-order correlation between ResNets of varying depth for three selection methods on CIFAR10 (and CIFAR100). For greedy k-centers, we started with 1,000 randomly selected points and ranked the remaining points based on the order they are added to set $s$ in Algorithm 1. Across models, there was a positive correlation similar to the correlation between runs of the same model. For forgetting events and entropy, we ranked points in descending order based on the number of forgetting events and the entropy of the output predictions from the trained model, respectively. Both metrics had comparable positive correlations between different models and different runs of the same model. We also looked at the Pearson correlation coefficient for the number of forgetting events and entropy in Figure 15 in the Appendix and found a similar positive correlation. The consistent positive correlation between varying depths illustrates why small models are good proxies for larger models in data selection.

---

[2] https://github.com/facebookresearch/fastText

**Models with different architectures.** We further investigated different model architectures by calculating the Spearman's correlation between pretrained ImageNet models and found that correlations were high across a wide range of models (Figure 8 in the Appendix). For example, MobileNet V2's (Sandler et al., 2018) entropy-based rankings were highly correlated to ResNet50 (on par with ResNet18), even though the model had far fewer parameters (3.5M vs. 25.6M). In concert with our fastText and VDCNN results, the high correlations between different model architectures suggest that SVP might be widely applicable. While there are likely limits to how different architectures can be, there is a wide range of trade-offs between accuracy and computational complexity, even within a narrow spectrum of models.

## 4    RELATED WORK

**Active learning.** There are examples in the active learning literature that address the computational efficiency of active learning methods by using one model to select points for a different, more expensive model. For instance, Lewis & Catlett (1994) proposed heterogeneous uncertainty sampling and used a Naïve Bayes classifier to select points to label for a more expensive decision tree target model. Tomanek et al. (2007) uses a committee-based active learning algorithm for an NLP task and notes that the set of selected points are "reusable" across different models (maximum entropy, conditional random field, naive Bayes). In our work, we showed that this can be generalized to deep learning by either using smaller models or fewer training epochs, where it can significantly reduce the running time of uncertainty-based (Settles, 2012; Shen et al., 2017; Gal et al., 2017) and recent representativeness-based (Sener & Savarese, 2018) methods.

**Core-set selection.** Core-set selection attempts to find a representative subset of points to speed up learning or clustering; such as $k$-means and $k$-medians (Har-Peled & Kushal, 2007), SVM (Tsang et al., 2005), Bayesian logistic regression (Huggins et al., 2016), and Bayesian inference (Campbell & Broderick, 2017; 2018). However, these examples generally require ready-to-use features as input, and do not directly apply to deep neural networks unless a feature representation is first learned, which usually requires training the full target model itself. There is also a body of work on data summarization based on submodular maximization (Wei et al., 2013; 2014; Tschiatschek et al., 2014; Ni et al., 2015), but these techniques depend on a combination of hand-engineered features and simple models (e.g., hidden Markov models and Gaussian mixture models) pretrained on auxiliary tasks. In comparison, our work demonstrated that we can use the feature representations of smaller, faster-to-train proxy models as an effective way to select core-sets for deep learning tasks.

Recently, Toneva et al. (2019) showed that a large number of "unforgettable" examples that are rarely incorrectly classified once learned (i.e., 30% on CIFAR10) could be omitted without impacting generalization, which can be viewed as a core-set selection method. They also provide initial evidence that forgetting events are transferable across models and throughout training by using the forgetting events from ResNet18 to select a subset for WideResNet (Zagoruyko & Komodakis, 2016) and by computing the Spearman's correlation of forgetting events during training compared to their final values. In our work, we evaluated a similar idea of using proxy models to approximate various properties of a large model, and showed that proxy models closely match the rankings of large models in the entropy, greedy k-centers, and example forgetting metrics.

## 5    CONCLUSION

In this work, we introduced selection via proxy (SVP) to improve the computational efficiency of active learning and core-set selection in deep learning by substituting a cheaper proxy model's representation for an expensive model's during data selection. Applied to least confidence uncertainty sampling and Sener & Savarese (2018)'s greedy k-centers approach, SVP achieved up to a $41.9\times$ and $3.8\times$ improvement in runtime respectively with no significant increase in error (often within 0.1%). For core-set selection, we found that SVP can remove up to 50% of the data from CIFAR10 in $10\times$ less time than it takes to train the target model, achieving a $1.6\times$ speed-up in end-to-end training.

---

Code available at `https://github.com/stanford-futuredata/selection-via-proxy`

ACKNOWLEDGMENTS

This research was supported in part by affiliate members and other supporters of the Stanford DAWN project—Ant Financial, Facebook, Google, Infosys, NEC, and VMware—as well as Toyota Research Institute, Northrop Grumman, Amazon Web Services, Cisco, and the NSF under CAREER grant CNS-1651570. Jure Leskovec is a Chan Zuckerberg Biohub investigator. We also gratefully acknowledge the support of DARPA under Nos. FA865018C7880 (ASED), N660011924033 (MCS); ARO under Nos. W911NF-16-1-0342 (MURI), W911NF-16-1-0171 (DURIP); NSF under Nos. OAC-1835598 (CINES), OAC-1934578 (HDR), DGE-1656518 (GRFP), DGE-114747 (GRFP); SNSF under Nos. P2EZP2_172187; Stanford Data Science Initiative, Wu Tsai Neurosciences Institute, Chan Zuckerberg Biohub, JD.com, Amazon, Boeing, Docomo, Huawei, Hitachi, Observe, Siemens, UST Global. The U.S. Government is authorized to reproduce and distribute reprints for Governmental purposes notwithstanding any copyright notation thereon. Any opinions, findings, and conclusions or recommendations expressed in this material are those of the authors and do not necessarily reflect the views, policies, or endorsements, either expressed or implied, of SNSF, NSF, DARPA, NIH, ARO, or the U.S. Government.

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

# A APPENDIX

## A.1 CHOICE OF DATASETS

For our experimental evaluation in Section 3, we focused on classification because it is a widely studied task in active learning (Lewis & Gale, 1994; Lewis & Catlett, 1994; Settles, 2012; Sener & Savarese, 2018; Kirsch et al., 2019; Mussmann & Liang, 2018; Gal et al., 2017; Houlsby et al., 2011). While there are a few bespoke solutions for machine translation (Peris & Casacuberta, 2018) and named entity recognition (Shen et al., 2017), we wanted to compare against the broader body of active learning research. Many popular active learning methods like uncertainty sampling (e.g., entropy, least confidence, and max-margin) assume a single categorical probability distribution for each example, which makes it hard to adapt to other domains. Instead of tackling open challenges for active learning like machine translation, we applied SVP to many classification datasets; however, the simplicity of SVP means that the core ideas of this paper can be applied more broadly in future work.

Specifically, we performed experiments on three image classification datasets: CIFAR10, CIFAR100 (Krizhevsky & Hinton, 2009), and ImageNet (Russakovsky et al., 2015); and two text classification datasets: Amazon Review Polarity and Full (Zhang & LeCun, 2015; Zhang et al., 2015). While multiple tasks on roughly the same data distribution may seem redundant, the data efficiency of active learning depends on error (Mussmann & Liang, 2018). We included both CIFAR10 (low-error) and CIFAR100 (high-error) to demonstrate that our approach performs as well as standard active learning at different points on the error and data efficiency curve. The same rationale is also valid for Amazon Review Polarity (low-error) and Full (high-error). However, the Amazon Review dataset adds a medium (text) and a much larger scale (3.6M and 3M examples, respectively). Adding ImageNet further allows us to investigate scale in the number of examples, but also the number of classes and the dimension of the input. To the best of our knowledge, we are the first active learning paper to present results on the full ImageNet classification task.

## A.2 IMPLEMENTATION DETAILS

**CIFAR10 and CIFAR100.** We used ResNet164 with pre-activation from He et al. (2016b) as our large target model for both CIFAR10 and CIFAR100. Note that as originally proposed in He et al. (2016a), the smaller, proxy models are also ResNet architectures with pre-activation, but they use pairs of $3 \times 3$ convolutional layers as their residual unit rather than bottlenecks and achieve lower accuracy as shown in Figure 4. As with He et al. (2016b), the ResNets we used were much narrower when applied to CIFAR rather than ImageNet (256 filters rather than 2048 in the final layer of the last bottleneck) and have fewer sections, which means far fewer weights despite the increased depth. For example, ResNet50 on ImageNet has ~25M weights while ResNet164 on CIFAR has ~1.7M (see Table 2). More recent networks such as Wide Residual Networks (Zagoruyko & Komodakis, 2016), ResNeXt (Xie et al., 2017), and DenseNets (Huang et al., 2017) use models with more than 25M parameters on CIFAR10, making ResNet164 relatively small in comparison. Core-set selection experiments used a single Nvidia P100 GPU, while the active learning experiments used a Titan V GPU. We followed the same training procedure, initialization, and hyperparameters as He et al. (2016b) with the exception of weight decay, which was set to 0.0005 and decreased the model's validation error in all conditions.

**ImageNet.** we used the original ResNet architecture from He et al. (2016a) implemented in PyTorch [3] (Paszke et al., 2017) with ResNet50 as the target and ResNet18 as the proxy. For training, we used a custom machine with 4 Nvidia Titan V GPUs and followed Nvidia's optimized implementation [4] with a larger batch size, appropriately scaled learning rate (Goyal et al., 2017), a 5-epoch warm-up period, and mixed precision training (Micikevicius et al., 2017) with the apex [5] library. For active learning, we used the same batch size of 768 images for both ResNet18 and ResNet50 for simplicity, which was the maximum batch size that could fit into memory for ResNet50. However, ResNet18 with a batch size of 768 underutilized the GPU and yielded a lower speed-up. With separate batch sizes for ResNet18 and ResNet50, we would have seen speed-ups closer to $2.7\times$.

---

[3] https://pytorch.org/docs/stable/torchvision/models.html
[4] https://github.com/NVIDIA/DeepLearningExamples
[5] https://github.com/NVIDIA/apex/tree/master/examples/imagenet

Table 2: Number of parameters in each model.

| Dataset | Model | Number of Parameters (millions) |
|---|---|---:|
| CIFAR10 | ResNet164 | 1.7 |
| | ResNet110 | 1.73 |
| | ResNet56 | 0.86 |
| | ResNet20 | 0.27 |
| | ResNet14 | 0.18 |
| | ResNet8 | 0.08 |
| CIFAR100 | ResNet164 | 1.73 |
| | ResNet110 | 1.74 |
| | ResNet56 | 0.86 |
| | ResNet20 | 0.28 |
| | ResNet14 | 0.18 |
| | ResNet8 | 0.08 |
| ImageNet | ResNet50 | 25.56 |
| | ResNet18 | 11.69 |
| Amazon Review Polarity | VDCNN29 | 16.64 |
| | VDCNN9 | 14.17 |
| Amazon Review Full | VDCNN29 | 16.64 |
| | VDCNN9 | 14.18 |

**Amazon Review Polarity (2-classes) and Full (5-classes).** For Amazon Review Polarity and Amazon Review Full, we used VDCNN (Conneau et al., 2017) and fastText (Joulin et al., 2016) with VDCNN29 as the target and fastText and VDCNN9 as proxies. For Amazon Review Polarity, core-set selection experiments used a single Nvidia P100 GPU, while the active learning experiments used a Nvidia Titan V GPU to train VDCNN models. For Amazon Review Full, core-set selection and active learning experiments both used a Nvidia Titan V GPU. In all settings, we used the same training procedure from Conneau et al. (2017) for VDCNN9 and VDCNN29. For fastText, we used Facebook's implementation [6] and followed the same training procedure from Joulin et al. (2016).

### A.3 MOTIVATION FOR CREATING PROXIES

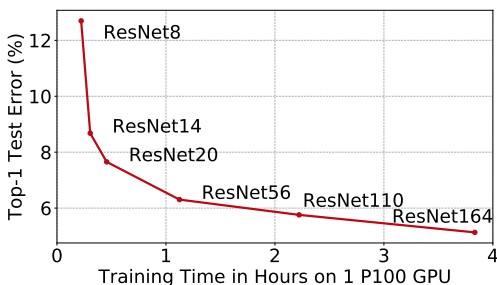
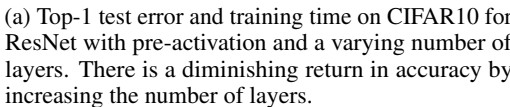
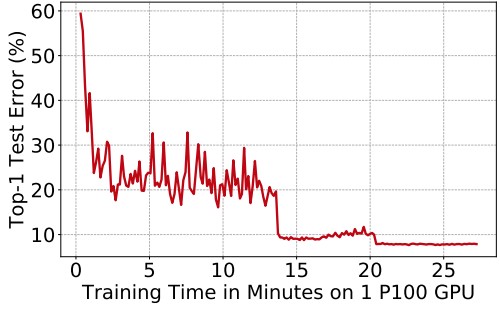

(a) Top-1 test error and training time on CIFAR10 for ResNet with pre-activation and a varying number of layers. There is a diminishing return in accuracy by increasing the number of layers.

(b) Top-1 test error during training of ResNet20 with pre-activation. In the first 14 minutes, ResNet20 reaches 9.0% top-1 error, while the remaining 12 minutes are spent on decreasing error to 7.6%

Figure 4: Top-1 test error on CIFAR10 for varying model sizes (left) and over the course of training a single model (right), demonstrating a large amount of time is spent on small changes in accuracy.

---

[6] https://github.com/facebookresearch/fastText

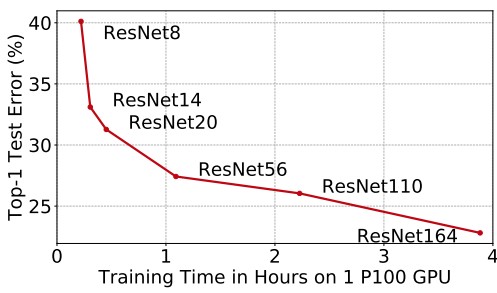 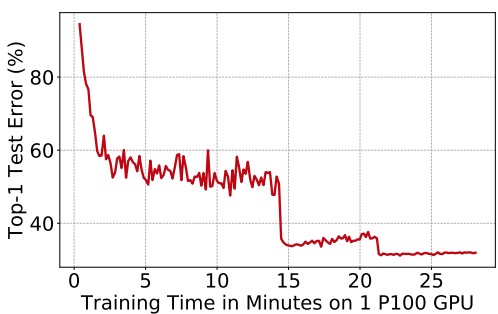

(a) Top-1 test error and training time on CIFAR100 for ResNet with pre-activation and a varying number of layers. There are diminishing returns in accuracy from increasing the number of layers.

(b) Top-1 test error during training of ResNet20 with pre-activation. In the first 15 minutes, ResNet20 reaches 33.9% top-1 error, while the remaining 12 minutes are spent on decreasing error to 31.1%

Figure 5: Top-1 test error on CIFAR100 for varying model sizes (left) and over the course of training a single model (right), demonstrating a large amount of time is spent on small changes in accuracy.

### A.4 ADDITIONAL ACTIVE LEARNING RESULTS

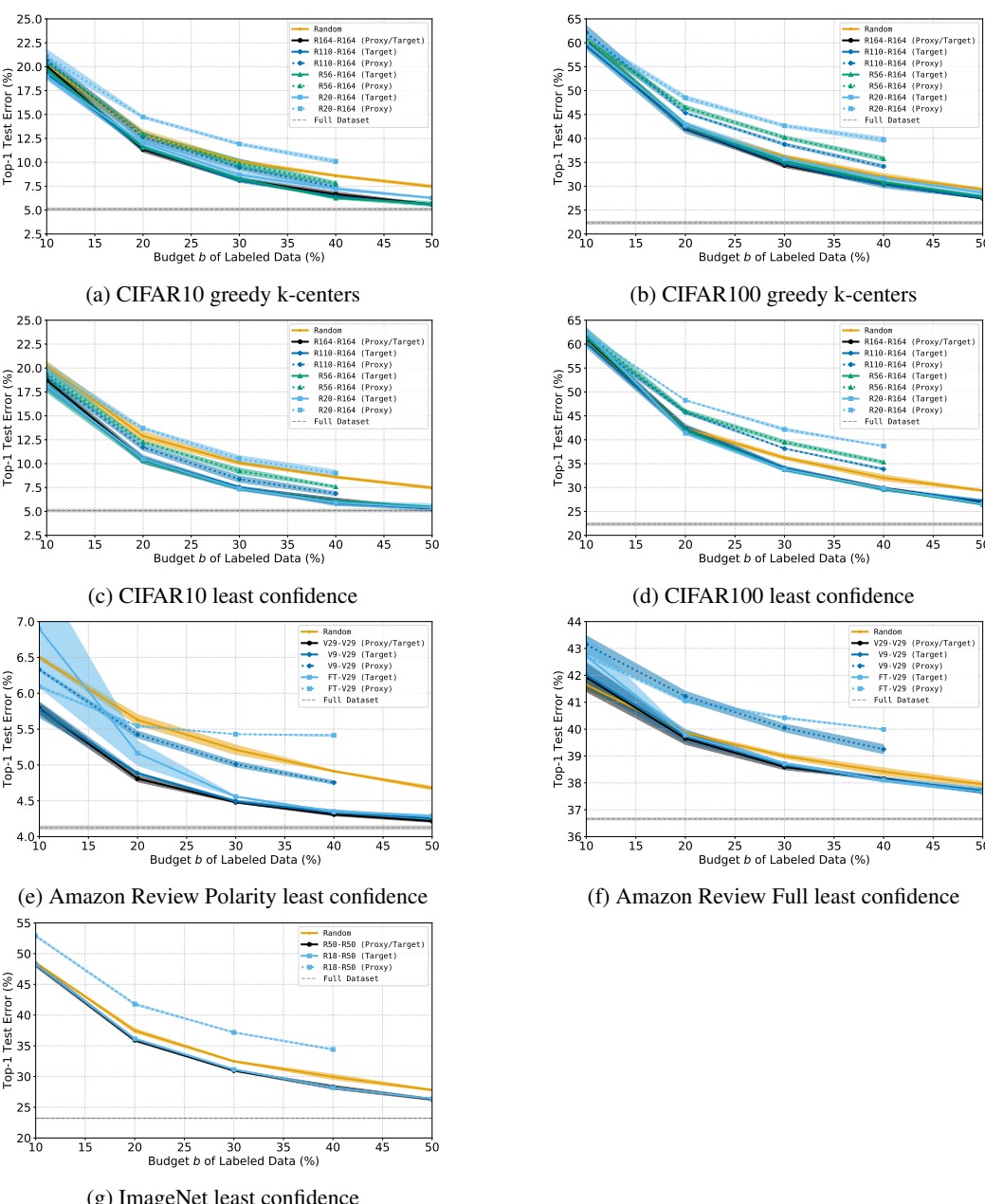

(a) CIFAR10 greedy k-centers

(b) CIFAR100 greedy k-centers

(c) CIFAR10 least confidence

(d) CIFAR100 least confidence

(e) Amazon Review Polarity least confidence

(f) Amazon Review Full least confidence

(g) ImageNet least confidence

Figure 6: **Quality of proxies compared to target models.** Average ($\pm$ 1 std.) top-1 error from 3 runs of active learning with varying proxies, selection methods, and budgets on five classification datasets. Dotted lines show the top-1 error of the proxy models, while solid lines show the top-1 error of the target models. CIFAR10 and CIFAR100 experiments used varying depths of pre-activation ResNet (R) models as proxies and ResNet164 (R164) as the target model (e.g., R20-R164 is ResNet20 selecting for ResNet164). ImageNet used ResNet18 (R18) as the proxy and ResNet50 (R50) as the target. Amazon Review Polarity and Amazon Review Full used VDCNN9 (V9) and fastText (FT) as proxies and VDCNN29 (V29) as the target. Across datasets, proxies, methods, and budgets, smaller proxies had higher top-1 error than the target model, but selecting points that were nearly as good as the points selected by the target that did not harm the final target model's predictive performance.

Table 3: **SVP performance on active learning.** Average ($\pm$ 1 std.) top-1 error and data selection speed-ups from 3 runs of active learning with varying proxies, methods, and labeling budgets on five datasets. **Bold** speed-ups indicate settings that either achieve lower error or are within 1 std. of the mean top-1 error for the baseline approach of using the same model for selection and the final predictions. Across datasets and methods, SVP sped up selection without significantly increasing the error of the final target.

| Dataset | Method | Selection Model | Top-1 Error of Target Model (%) | | | | | Data Selection Speed-up | | | | |
|---|---|---|---|---|---|---|---|---|---|---|---|---|
| | | Budget ($b/n$) | 10% | 20% | 30% | 40% | 50% | 10% | 20% | 30% | 40% | 50% |
| CIFAR10 | Random | - | 20.3 ± 0.51 | 12.9 ± 0.37 | 10.1 ± 0.24 | 8.5 ± 0.22 | 7.5 ± 0.11 | - | - | - | - | - |
| | Least Confidence | ResNet164 (Baseline) | 18.7 ± 0.31 | 10.4 ± 0.38 | 7.4 ± 0.16 | 6.1 ± 0.32 | 5.3 ± 0.06 | **1.0×** | **1.0×** | **1.0×** | **1.0×** | **1.0×** |
| | | ResNet110 | 18.1 ± 0.41 | 10.5 ± 0.06 | 7.5 ± 0.11 | 5.9 ± 0.33 | 5.3 ± 0.05 | **1.8×** | **1.9×** | **1.9×** | **1.8×** | **1.8×** |
| | | ResNet56 | 18.2 ± 0.73 | 10.3 ± 0.28 | 7.4 ± 0.10 | 6.1 ± 0.06 | 5.5 ± 0.08 | **2.6×** | **2.9×** | **3.0×** | **3.1×** | **3.1×** |
| | | ResNet20 | 18.1 ± 0.28 | 10.5 ± 0.42 | 7.4 ± 0.23 | 5.9 ± 0.19 | 5.4 ± 0.41 | **3.8×** | **5.8×** | **6.7×** | **7.0×** | **7.2×** |
| | Greedy k-Centers | ResNet164 (Baseline) | 20.1 ± 0.39 | 11.3 ± 0.40 | 8.1 ± 0.22 | 6.6 ± 0.24 | 5.6 ± 0.04 | **1.0×** | **1.0×** | **1.0×** | **1.0×** | **1.0×** |
| | | ResNet110 | 19.4 ± 0.55 | 11.6 ± 0.16 | 8.1 ± 0.16 | 6.4 ± 0.10 | 5.7 ± 0.13 | **2.1×** | **1.8×** | **1.7×** | **1.7×** | **1.6×** |
| | | ResNet56 | 19.8 ± 0.49 | 11.6 ± 0.16 | 8.4 ± 0.21 | 6.3 ± 0.17 | 5.7 ± 0.19 | **3.0×** | **2.9×** | **2.8×** | **2.8×** | **2.8×** |
| | | ResNet20 | 19.5 ± 0.76 | 12.1 ± 0.44 | 8.8 ± 0.31 | 7.2 ± 0.19 | 6.1 ± 0.18 | 3.8× | 4.6× | 5.0× | 5.3× | 5.5× |
| CIFAR100 | Random | - | 60.7 ± 0.81 | 42.5 ± 0.55 | 36.0 ± 0.42 | 31.9 ± 0.48 | 29.3 ± 0.16 | - | - | - | - | - |
| | Least Confidence | ResNet164 (Baseline) | 61.2 ± 1.09 | 42.2 ± 0.67 | 33.9 ± 0.33 | 29.9 ± 0.18 | 26.9 ± 0.21 | **1.0×** | **1.0×** | **1.0×** | **1.0×** | **1.0×** |
| | | ResNet110 | 60.2 ± 0.84 | 42.3 ± 0.95 | 34.1 ± 0.38 | 29.7 ± 0.41 | 27.2 ± 0.25 | **1.5×** | **1.6×** | **1.6×** | **1.6×** | 1.6× |
| | | ResNet56 | 61.5 ± 0.93 | 42.0 ± 0.17 | 33.7 ± 0.33 | 29.7 ± 0.08 | 26.4 ± 0.13 | **2.4×** | **2.7×** | **3.0×** | **2.9×** | **3.1×** |
| | | ResNet20 | 62.4 ± 1.07 | 41.4 ± 0.25 | 33.8 ± 0.37 | 29.8 ± 0.10 | 26.6 ± 0.14 | **4.0×** | **5.8×** | **6.6×** | **7.0×** | **7.2×** |
| | Greedy k-Centers | ResNet164 (Baseline) | 60.4 ± 1.30 | 42.4 ± 0.57 | 34.5 ± 0.40 | 30.2 ± 0.33 | 27.3 ± 0.24 | **1.0×** | **1.0×** | **1.0×** | **1.0×** | **1.0×** |
| | | ResNet110 | 59.6 ± 0.78 | 42.2 ± 0.76 | 34.9 ± 0.40 | 30.3 ± 0.46 | 27.4 ± 0.21 | **2.3×** | **1.9×** | **1.8×** | **1.7×** | **1.6×** |
| | | ResNet56 | 60.9 ± 1.08 | 42.6 ± 0.47 | 35.2 ± 0.40 | 30.8 ± 0.25 | 27.8 ± 0.23 | **3.3×** | **3.2×** | 3.1× | 3.1× | 3.0× |
| | | ResNet20 | 60.2 ± 1.27 | 42.9 ± 0.52 | 35.8 ± 0.45 | 31.6 ± 0.31 | 28.5 ± 0.48 | **4.5×** | **5.5×** | 5.9× | 6.1× | 6.2× |
| ImageNet | Random | - | 48.5 ± 0.04 | 37.5 ± 0.34 | 32.5 ± 0.12 | 29.9 ± 0.42 | 27.8 ± 0.13 | - | - | - | - | - |
| | Least Confidence | ResNet50 (Baseline) | 48.2 ± 0.37 | 35.9 ± 0.22 | 31.0 ± 0.10 | 28.3 ± 0.32 | 26.3 ± 0.16 | **1.0×** | **1.0×** | **1.0×** | **1.0×** | **1.0×** |
| | | ResNet18 | 48.3 ± 0.31 | 36.1 ± 0.19 | 31.1 ± 0.12 | 28.2 ± 0.13 | 26.4 ± 0.02 | **1.2×** | **1.3×** | **1.4×** | **1.5×** | **1.6×** |
| Amazon Review Polarity | Random | - | 6.5 ± 0.03 | 5.6 ± 0.07 | 5.2 ± 0.07 | 4.9 ± 0.01 | 4.7 ± 0.03 | - | - | - | - | - |
| | Least Confidence | VDCNN29 (Baseline) | 5.8 ± 0.08 | 4.8 ± 0.04 | 4.5 ± 0.01 | 4.3 ± 0.02 | 4.2 ± 0.02 | **1.0×** | **1.0×** | **1.0×** | **1.0×** | **1.0×** |
| | | VDCNN9 | 5.8 ± 0.11 | 4.9 ± 0.01 | 4.5 ± 0.02 | 4.3 ± 0.04 | 4.3 ± 0.03 | **1.9×** | 1.8× | **1.8×** | **1.8×** | 1.8× |
| | | fastText | 6.9 ± 0.81 | 5.2 ± 0.17 | 4.6 ± 0.01 | 4.3 ± 0.01 | 4.3 ± 0.02 | 10.6× | 20.6× | 32.2× | **41.9×** | 51.3× |
| Amazon Review Full | Random | - | 41.7 ± 0.19 | 39.9 ± 0.05 | 39.0 ± 0.09 | 38.4 ± 0.14 | 37.9 ± 0.10 | - | - | - | - | - |
| | Least Confidence | VDCNN29 (Baseline) | 41.9 ± 0.54 | 39.7 ± 0.22 | 38.6 ± 0.10 | 38.2 ± 0.03 | 37.6 ± 0.01 | **1.0×** | **1.0×** | **1.0×** | **1.0×** | **1.0×** |
| | | VDCNN9 | 42.0 ± 0.44 | 39.8 ± 0.23 | 38.7 ± 0.09 | 38.1 ± 0.09 | 37.7 ± 0.10 | **2.0×** | **1.9×** | **1.8×** | **1.8×** | **1.8×** |
| | | fastText | 42.7 ± 0.77 | 39.8 ± 0.02 | 38.7 ± 0.05 | 38.1 ± 0.06 | 37.7 ± 0.05 | **8.7×** | **17.7×** | **26.7×** | **35.1×** | **43.1×** |

Table 4: **Performance of training for fewer epochs on active learning.** Average (± 1 std.) top-1 error and data selection speed-ups from 3 runs of active learning with varying proxies trained for a varying number of epochs on CIFAR10, CIFAR100, and ImageNet. **Bold** speed-ups indicate settings that either achieve lower error or are within 1 std. of the mean top-1 error for the baseline approach of using the same model for selection and the final predictions. Training for fewer epochs can provide a significant improvement over random sampling but is not quite as good as training for the full schedule.

| Dataset | Method | Selection Model | Budget ($b/n$) Epochs | Top-1 Error of Target Model (%) 10% | 20% | 30% | 40% | 50% | Data Selection Speed-up 10% | 20% | 30% | 40% | 50% |
|---|---|---|---|---|---|---|---|---|---|---|---|---|---|
| CIFAR10 | Random | - | - | 20.2 ± 0.49 | 12.9 ± 0.50 | 10.1 ± 0.18 | 8.6 ± 0.12 | 7.5 ± 0.15 | - | - | - | - | - |
| | Least Confidence | ResNet164 (Baseline) | 181 | 18.7 ± 0.31 | 10.4 ± 0.38 | 7.4 ± 0.16 | 6.1 ± 0.32 | 5.3 ± 0.06 | **1.0×** | **1.0×** | **1.0×** | **1.0×** | **1.0×** |
| | | | 100 | 18.4 ± 0.23 | 10.3 ± 0.13 | 7.3 ± 0.08 | 6.0 ± 0.16 | 5.3 ± 0.13 | **1.8×** | **1.8×** | **1.8×** | **1.8×** | **1.8×** |
| | | | 50 | 19.2 ± 0.35 | 11.7 ± 0.55 | 8.4 ± 0.15 | 7.2 ± 0.15 | 5.9 ± 0.14 | 3.4× | 3.6× | 3.6× | 3.6× | 3.6× |
| | | ResNet20 | 181 | 18.1 ± 0.28 | 10.5 ± 0.42 | 7.4 ± 0.23 | 5.9 ± 0.19 | 5.4 ± 0.41 | **3.8×** | **5.8×** | **6.7×** | **7.0×** | 7.2× |
| | | | 100 | 18.4 ± 0.38 | 10.3 ± 0.20 | 7.4 ± 0.13 | 5.9 ± 0.31 | 5.3 ± 0.16 | **6.8×** | **10.3×** | **11.6×** | **12.3×** | **12.7×** |
| | | | 50 | 18.8 ± 0.81 | 11.5 ± 0.33 | 8.5 ± 0.19 | 6.8 ± 0.09 | 5.9 ± 0.31 | **11.4×** | 19.4× | 23.1× | 24.7× | 25.6× |
| | Greedy k-Centers | ResNet164 (Baseline) | 181 | 20.1 ± 0.38 | 11.3 ± 0.26 | 8.2 ± 0.19 | 6.7 ± 0.25 | 5.6 ± 0.05 | **1.0×** | **1.0×** | **1.0×** | **1.0×** | **1.0×** |
| | | | 100 | 19.9 ± 0.41 | 11.9 ± 0.08 | 8.7 ± 0.22 | 6.8 ± 0.10 | 6.1 ± 0.10 | 1.3× | 1.4× | 1.4× | 1.5× | 1.5× |
| | | | 50 | 21.9 ± 0.58 | 13.3 ± 0.23 | 9.9 ± 0.29 | 8.0 ± 0.20 | 7.0 ± 0.11 | 1.6× | 1.9× | 2.2× | 2.3× | 2.5× |
| | | ResNet110 | 181 | 19.3 ± 0.51 | 11.5 ± 0.21 | 8.1 ± 0.19 | 6.4 ± 0.07 | 5.6 ± 0.10 | **2.1×** | **1.8×** | **1.6×** | **1.6×** | **1.5×** |
| | | | 100 | 19.6 ± 0.74 | 11.9 ± 0.15 | 8.6 ± 0.14 | 6.9 ± 0.06 | 5.8 ± 0.07 | 3.2× | 2.8× | 2.6× | 2.6× | 2.5× |
| | | | 50 | 21.0 ± 0.91 | 12.9 ± 0.31 | 9.7 ± 0.31 | 8.0 ± 0.12 | 6.9 ± 0.12 | 4.7× | 4.8× | 4.7× | 4.7× | 4.7× |
| | | ResNet56 | 181 | 19.7 ± 0.48 | 11.6 ± 0.20 | 8.3 ± 0.18 | 6.2 ± 0.10 | 5.7 ± 0.24 | **2.9×** | **2.7×** | **2.6×** | **2.7×** | **2.6×** |
| | | | 100 | 19.7 ± 0.83 | 11.7 ± 0.26 | 8.7 ± 0.23 | 7.0 ± 0.19 | 6.2 ± 0.17 | 4.6× | 4.7× | 4.8× | 4.8× | 4.8× |
| | | | 50 | 20.5 ± 0.05 | 13.0 ± 0.44 | 9.6 ± 0.26 | 8.1 ± 0.19 | 7.0 ± 0.15 | 6.3× | 7.5× | 8.1× | 8.4× | 8.8× |
| | | ResNet20 | 181 | 19.0 ± 0.35 | 12.0 ± 0.61 | 8.7 ± 0.32 | 7.2 ± 0.19 | 6.3 ± 0.06 | **3.7×** | **4.5×** | **4.8×** | **5.0×** | **5.2×** |
| | | | 100 | 20.1 ± 0.31 | 12.6 ± 0.19 | 9.1 ± 0.08 | 7.4 ± 0.35 | 6.4 ± 0.11 | 5.8× | 7.6× | 8.9× | 9.7× | 10.2× |
| | | | 50 | 21.6 ± 0.55 | 13.7 ± 0.22 | 10.4 ± 0.04 | 8.3 ± 0.23 | 7.1 ± 0.15 | 7.7× | 11.1× | 13.7× | 15.6× | 17.2× |
| CIFAR100 | Random | - | - | 61.0 ± 0.31 | 42.3 ± 0.61 | 36.2 ± 0.36 | 32.0 ± 0.61 | 29.4 ± 0.20 | - | - | - | - | - |
| | Least Confidence | ResNet164 (Baseline) | 181 | 61.2 ± 1.09 | 42.2 ± 0.67 | 33.9 ± 0.33 | 29.9 ± 0.18 | 26.9 ± 0.21 | **1.0×** | **1.0×** | **1.0×** | **1.0×** | **1.0×** |
| | | | 100 | 61.1 ± 1.57 | 41.4 ± 0.27 | 33.8 ± 0.60 | 29.8 ± 0.21 | 26.9 ± 0.24 | **1.8×** | **1.8×** | **1.8×** | **1.8×** | **1.8×** |
| | | | 50 | 62.5 ± 2.49 | 44.1 ± 0.24 | 35.2 ± 0.23 | 30.8 ± 0.47 | 27.5 ± 0.43 | 3.3× | 3.6× | 3.6× | 3.6× | 3.6× |
| | | ResNet20 | 181 | 62.4 ± 1.07 | 41.4 ± 0.25 | 33.8 ± 0.37 | 29.8 ± 0.10 | 26.6 ± 0.14 | **4.0×** | **5.8×** | **6.6×** | **7.0×** | 7.2× |
| | | | 100 | 61.9 ± 1.42 | 42.2 ± 0.61 | 34.3 ± 0.37 | 29.7 ± 0.06 | 27.0 ± 0.26 | **6.8×** | **10.4×** | **12.0×** | **12.7×** | **13.1×** |
| | | | 50 | 62.7 ± 1.40 | 43.5 ± 0.58 | 35.4 ± 0.23 | 30.9 ± 0.23 | 27.9 ± 0.64 | **11.5×** | 18.9× | 22.4× | 24.3× | 25.1× |
| | Greedy k-Centers | ResNet164 (Baseline) | 181 | 60.5 ± 0.90 | 42.1 ± 0.47 | 34.4 ± 0.45 | 30.4 ± 0.30 | 27.4 ± 0.05 | **1.0×** | **1.0×** | **1.0×** | **1.0×** | **1.0×** |
| | | | 100 | 60.4 ± 1.65 | 42.7 ± 0.50 | 34.9 ± 0.09 | 30.3 ± 0.46 | 27.8 ± 0.43 | 1.3× | 1.4× | 1.5× | 1.5× | 1.6× |
| | | | 50 | 60.5 ± 0.17 | 43.0 ± 0.11 | 36.3 ± 0.33 | 32.2 ± 0.22 | 29.2 ± 0.23 | 1.7× | 2.0× | 2.3× | 2.5× | 2.6× |
| | | ResNet110 | 181 | 59.3 ± 0.70 | 42.2 ± 1.06 | 34.7 ± 0.40 | 30.3 ± 0.65 | 27.5 ± 0.17 | **2.3×** | **1.9×** | **1.8×** | **1.7×** | **1.6×** |
| | | | 100 | 60.1 ± 0.60 | 42.7 ± 0.47 | 35.0 ± 0.20 | 30.6 ± 0.38 | 27.7 ± 0.12 | 3.1× | 2.7× | 2.5× | 2.4× | 2.3× |
| | | | 50 | 62.6 ± 1.54 | 43.8 ± 0.51 | 36.4 ± 0.97 | 32.4 ± 0.25 | 29.1 ± 0.48 | 5.1× | 5.0× | 5.0× | 5.0× | 5.0× |
| | | ResNet56 | 181 | 60.7 ± 0.71 | 42.7 ± 0.47 | 35.3 ± 0.55 | 30.8 ± 0.33 | 27.9 ± 0.07 | **3.4×** | **3.2×** | **3.2×** | **3.1×** | **3.1×** |
| | | | 100 | 60.9 ± 1.07 | 43.2 ± 0.38 | 35.1 ± 0.38 | 30.9 ± 0.57 | 27.8 ± 0.06 | 4.6× | 4.7× | 4.7× | 4.9× | 5.0× |
| | | | 50 | 60.1 ± 1.27 | 44.4 ± 0.56 | 36.5 ± 0.32 | 32.6 ± 0.62 | 29.6 ± 0.53 | 6.8× | 8.1× | 8.7× | 9.1× | 9.2× |
| | | ResNet20 | 181 | 60.0 ± 0.59 | 42.8 ± 0.70 | 35.8 ± 0.63 | 31.7 ± 0.40 | 28.7 ± 0.54 | **4.5×** | **5.4×** | **5.8×** | **5.9×** | **6.1×** |
| | | | 100 | 61.9 ± 1.01 | 43.2 ± 0.33 | 35.6 ± 0.23 | 31.4 ± 0.38 | 28.8 ± 0.23 | 6.4× | 8.2× | 9.5× | 10.3× | 10.9× |
| | | | 50 | 61.6 ± 0.65 | 45.1 ± 0.61 | 37.3 ± 1.05 | 32.9 ± 0.49 | 30.2 ± 0.18 | 8.1× | 11.6× | 14.3× | 16.3× | 17.7× |
| ImageNet | Random | - | - | 48.5 ± 0.04 | 37.5 ± 0.34 | 32.5 ± 0.12 | 29.9 ± 0.42 | 27.8 ± 0.13 | - | - | - | - | - |
| | Least Confidence | ResNet50 (Baseline) | 90 | 48.2 ± 0.37 | 35.9 ± 0.22 | 31.0 ± 0.10 | 28.3 ± 0.32 | 26.3 ± 0.16 | **1.0×** | **1.0×** | **1.0×** | **1.0×** | **1.0×** |
| | | | 45 | 48.7 ± 0.21 | 36.3 ± 0.03 | 31.3 ± 0.02 | 28.3 ± 0.19 | 26.5 ± 0.17 | 1.7× | 1.8× | 1.8× | **1.8×** | 1.7× |
| | | ResNet18 | 90 | 48.3 ± 0.31 | 36.1 ± 0.19 | 31.1 ± 0.12 | 28.2 ± 0.13 | 26.4 ± 0.02 | **1.2×** | **1.3×** | **1.4×** | **1.5×** | **1.6×** |
| | | | 45 | 48.3 ± 0.31 | 36.3 ± 0.07 | 31.3 ± 0.02 | 28.4 ± 0.17 | 26.6 ± 0.08 | 2.1× | 2.5× | 2.7× | **2.9×** | 3.1× |

## A.5 Additional Core-Set Selection Results

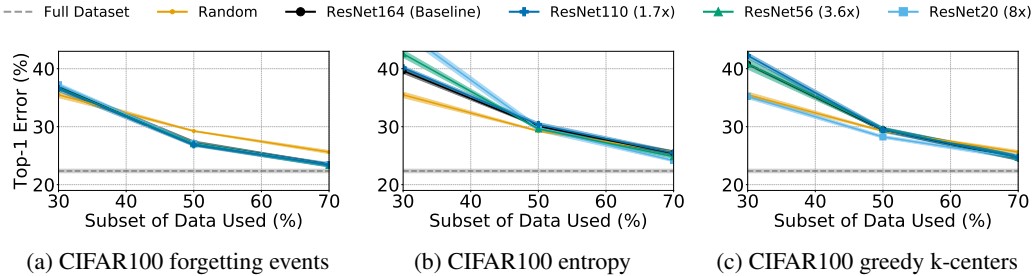

(a) CIFAR100 forgetting events     (b) CIFAR100 entropy     (c) CIFAR100 greedy k-centers

Figure 7: **SVP performance on core-set selection.** Average ($\pm$ 1 std.) top-1 error of ResNet164 over 5 runs of core-set selection with different selection methods, proxies, and subset sizes on CIFAR100. We found subsets using forgetting events (left), entropy (middle), and greedy k-centers (right) from a proxy model trained over the entire dataset. Across datasets and selection methods, SVP performed as well as an oracle baseline but significantly faster (speed-ups in parentheses).

Table 5: Average top-1 error ($\pm$ 1 std.) from 3 runs of core-set selection with varying selection methods on ImageNet, Amazon Review Polarity, and Amazon Review Full.

| | | | Top-1 Error (%) | | | |
|---|---|---|---|---|---|---|
| Dataset | Method | Selection Model | Subset Size 40% | 60% | 80% | 100% |
| ImageNet | Random | - | $32.2 \pm 0.12$ | $28.0 \pm 0.15$ | $25.8 \pm 0.06$ | $23.3 \pm 0.11$ |
| | Entropy | ResNet50 (Baseline) | $34.9 \pm 0.08$ | $28.8 \pm 0.03$ | $25.9 \pm 0.04$ | - |
| | Entropy | ResNet18 | $32.2 \pm 0.04$ | $27.0 \pm 0.01$ | $25.1 \pm 0.07$ | - |
| | Forgetting Events | ResNet50 (Baseline) | $31.9 \pm 0.07$ | $26.7 \pm 0.06$ | $24.8 \pm 0.03$ | - |
| | Forgetting Events | ResNet18 | $31.6 \pm 0.07$ | $27.1 \pm 0.10$ | $25.3 \pm 0.18$ | - |
| Amazon Review Polarity | Random | - | $4.9 \pm 0.02$ | $4.5 \pm 0.05$ | $4.3 \pm 0.01$ | $4.1 \pm 0.04$ |
| | Entropy | VDCNN29 (Baseline) | $4.4 \pm 0.03$ | $4.2 \pm 0.02$ | $4.2 \pm 0.02$ | - |
| | Entropy | VDCNN9 | $4.4 \pm 0.02$ | $4.2 \pm 0.01$ | $4.2 \pm 0.00$ | - |
| | Entropy | fastText | $4.4 \pm 0.02$ | $4.2 \pm 0.02$ | $4.2 \pm 0.02$ | - |
| Amazon Review Full | Random | - | $38.4 \pm 0.03$ | $37.6 \pm 0.03$ | $37.0 \pm 0.05$ | $36.6 \pm 0.06$ |
| | Entropy | VDCNN29 (Baseline) | $42.7 \pm 1.14$ | $39.3 \pm 0.14$ | $37.6 \pm 0.10$ | - |
| | Entropy | VDCNN9 | $41.1 \pm 0.24$ | $38.8 \pm 0.03$ | $37.7 \pm 0.09$ | - |
| | Entropy | fastText | $39.0 \pm 0.18$ | $37.8 \pm 0.06$ | $37.1 \pm 0.06$ | - |

Table 6: Average (± 1 std.) top-1 error and runtime in minutes from 5 runs of core-set selection with varying proxies, selection methods, and subset sizes on CIFAR10 and CIFAR100.

| Dataset | Method | Selection Model | Top-1 Error of ResNet164 (%) 30% | 50% | 70% | Data Selection Runtime in Minutes 30% | 50% | 70% | Total Runtime in Minutes 30% | 50% | 70% |
|---|---|---|---|---|---|---|---|---|---|---|---|
| CIFAR10 | Facility Location | ResNet164 (Baseline) | $8.9 \pm 0.29$ | $6.3 \pm 0.23$ | $5.4 \pm 0.09$ | $265 \pm 48.0$ | $286 \pm 91.6$ | $260 \pm 42.6$ | $342 \pm 47.7$ | $406 \pm 94.3$ | $425 \pm 41.7$ |
| | | ResNet20 | $9.1 \pm 0.33$ | $6.4 \pm 0.13$ | $5.5 \pm 0.21$ | $27 \pm 1.1$ | $28 \pm 1.4$ | $30 \pm 2.2$ | $104 \pm 1.9$ | $147 \pm 1.0$ | $193 \pm 5.7$ |
| | | ResNet56 | $8.9 \pm 0.09$ | $6.1 \pm 0.21$ | $5.3 \pm 0.07$ | $65 \pm 3.9$ | $67 \pm 3.8$ | $68 \pm 3.4$ | $142 \pm 4.7$ | $187 \pm 4.8$ | $230 \pm 5.1$ |
| | Forgetting Events | ResNet164 (Baseline) | $7.7 \pm 0.19$ | $5.2 \pm 0.11$ | $5.0 \pm 0.12$ | $218 \pm 1.4$ | $218 \pm 1.6$ | $219 \pm 1.5$ | $296 \pm 3.2$ | $340 \pm 6.8$ | $382 \pm 4.6$ |
| | | ResNet20 | $7.6 \pm 0.18$ | $5.2 \pm 0.11$ | $5.1 \pm 0.07$ | $24 \pm 1.3$ | $24 \pm 1.4$ | $25 \pm 1.5$ | $101 \pm 2.6$ | $142 \pm 2.5$ | $185 \pm 5.0$ |
| | | ResNet56 | $7.7 \pm 0.27$ | $5.2 \pm 0.09$ | $5.1 \pm 0.09$ | $63 \pm 4.3$ | $63 \pm 4.0$ | $63 \pm 4.0$ | $141 \pm 5.4$ | $184 \pm 4.6$ | $226 \pm 2.8$ |
| | Entropy | ResNet164 (Baseline) | $9.6 \pm 0.16$ | $6.4 \pm 0.27$ | $5.6 \pm 0.19$ | $218 \pm 1.4$ | $218 \pm 1.7$ | $218 \pm 1.6$ | $296 \pm 1.5$ | $338 \pm 2.2$ | $382 \pm 3.1$ |
| | | ResNet20 | $8.9 \pm 0.18$ | $5.7 \pm 0.23$ | $5.3 \pm 0.09$ | $24 \pm 1.3$ | $24 \pm 1.5$ | $25 \pm 1.5$ | $103 \pm 2.2$ | $145 \pm 1.3$ | $190 \pm 3.7$ |
| | | ResNet56 | $9.9 \pm 0.29$ | $6.6 \pm 0.09$ | $5.7 \pm 0.17$ | $63 \pm 4.3$ | $63 \pm 4.0$ | $63 \pm 4.0$ | $141 \pm 4.8$ | $182 \pm 4.0$ | $226 \pm 3.8$ |
| CIFAR100 | Facility Location | ResNet164 (Baseline) | $40.8 \pm 0.20$ | $29.5 \pm 0.29$ | $24.6 \pm 0.42$ | $325 \pm 52.2$ | $296 \pm 158.7$ | $296 \pm 70.2$ | $339 \pm 52.7$ | $446 \pm 158.1$ | $460 \pm 69.1$ |
| | | ResNet20 | $35.2 \pm 0.37$ | $28.2 \pm 0.23$ | $24.7 \pm 0.30$ | $27 \pm 0.8$ | $28 \pm 1.3$ | $30 \pm 1.4$ | $105 \pm 2.6$ | $151 \pm 3.6$ | $198 \pm 4.6$ |
| | | ResNet56 | $40.8 \pm 0.89$ | $29.6 \pm 0.28$ | $24.7 \pm 0.40$ | $64 \pm 1.7$ | $66 \pm 1.9$ | $67 \pm 2.2$ | $142 \pm 1.7$ | $185 \pm 1.5$ | $230 \pm 3.9$ |
| | | ResNet110 | $42.3 \pm 0.44$ | $29.5 \pm 0.43$ | $24.7 \pm 0.38$ | $129 \pm 3.7$ | $131 \pm 3.6$ | $132 \pm 3.5$ | $208 \pm 7.3$ | $253 \pm 8.5$ | $303 \pm 11.6$ |
| | Forgetting Events | ResNet164 (Baseline) | $36.8 \pm 0.36$ | $27.1 \pm 0.40$ | $23.5 \pm 0.19$ | $221 \pm 6.1$ | $221 \pm 6.1$ | $221 \pm 6.1$ | $298 \pm 5.7$ | $342 \pm 5.5$ | $384 \pm 4.7$ |
| | | ResNet20 | $37.2 \pm 0.29$ | $27.1 \pm 0.14$ | $23.4 \pm 0.16$ | $24 \pm 0.7$ | $25 \pm 0.7$ | $25 \pm 0.7$ | $104 \pm 3.3$ | $148 \pm 3.6$ | $193 \pm 6.1$ |
| | | ResNet56 | $36.7 \pm 0.23$ | $27.0 \pm 0.33$ | $23.3 \pm 0.28$ | $62 \pm 2.4$ | $62 \pm 2.6$ | $62 \pm 1.9$ | $141 \pm 7.1$ | $183 \pm 3.8$ | $228 \pm 5.2$ |
| | | ResNet110 | $36.6 \pm 0.51$ | $26.9 \pm 0.27$ | $23.4 \pm 0.37$ | $127 \pm 2.7$ | $127 \pm 2.7$ | $127 \pm 2.7$ | $207 \pm 3.7$ | $250 \pm 4.9$ | $293 \pm 7.3$ |
| | Entropy | ResNet164 (Baseline) | $39.6 \pm 0.43$ | $30.1 \pm 0.12$ | $25.4 \pm 0.39$ | $220 \pm 6.4$ | $220 \pm 6.4$ | $220 \pm 6.4$ | $297 \pm 7.3$ | $340 \pm 7.3$ | $380 \pm 7.1$ |
| | | ResNet20 | $46.5 \pm 0.74$ | $29.7 \pm 0.45$ | $24.2 \pm 0.21$ | $24 \pm 0.6$ | $25 \pm 0.7$ | $24 \pm 0.6$ | $105 \pm 1.7$ | $148 \pm 2.6$ | $193 \pm 3.6$ |
| | | ResNet56 | $42.6 \pm 0.63$ | $29.6 \pm 0.13$ | $24.8 \pm 0.29$ | $62 \pm 1.7$ | $62 \pm 1.8$ | $62 \pm 1.9$ | $142 \pm 1.9$ | $186 \pm 3.9$ | $230 \pm 5.9$ |
| | | ResNet110 | $40.2 \pm 0.28$ | $30.4 \pm 0.35$ | $25.5 \pm 0.34$ | $127 \pm 3.0$ | $127 \pm 3.1$ | $127 \pm 3.1$ | $204 \pm 3.3$ | $247 \pm 3.5$ | $291 \pm 3.7$ |

Table 7: Average top-1 error (± 1 std.) and runtime in minutes from 5 runs of core-set selection with varying selection methods calculated from ResNet20 models trained for a varying number of epochs on CIFAR10 and CIFAR100.

| Dataset | Method | Selection Model | Epochs | Top-1 Error of ResNet164 (%) 30% | 50% | 70% | Data Selection Runtime in Minutes 30% | 50% | 70% | Total Runtime in Minutes 30% | 50% | 70% |
|---|---|---|---|---|---|---|---|---|---|---|---|---|
| CIFAR10 | Forgetting Events | ResNet164 (Baseline) | 181 | $7.7 \pm 0.19$ | $5.2 \pm 0.11$ | $5.0 \pm 0.12$ | $218 \pm 1.4$ | $218 \pm 1.6$ | $219 \pm 1.5$ | $296 \pm 3.2$ | $340 \pm 6.8$ | $382 \pm 4.6$ |
| | | ResNet20 | 181 | $7.6 \pm 0.18$ | $5.2 \pm 0.11$ | $5.1 \pm 0.07$ | $24 \pm 1.4$ | $24 \pm 1.4$ | $25 \pm 1.5$ | $101 \pm 2.6$ | $142 \pm 2.5$ | $185 \pm 5.0$ |
| | | | 100 | $7.1 \pm 0.16$ | $5.4 \pm 0.22$ | $5.0 \pm 0.17$ | $14 \pm 1.0$ | $14 \pm 0.7$ | $14 \pm 0.7$ | $92 \pm 1.5$ | $135 \pm 0.7$ | $178 \pm 2.5$ |
| | | | 50 | $7.2 \pm 0.18$ | $5.4 \pm 0.09$ | $5.1 \pm 0.15$ | $7 \pm 0.9$ | $7 \pm 0.4$ | $7 \pm 0.4$ | $85 \pm 2.0$ | $126 \pm 1.4$ | $169 \pm 1.0$ |
| | | | 25 | $7.3 \pm 0.17$ | $5.4 \pm 0.09$ | $5.1 \pm 0.12$ | $4 \pm 0.4$ | $4 \pm 0.2$ | $4 \pm 0.2$ | $80 \pm 0.8$ | $122 \pm 1.5$ | $164 \pm 1.5$ |
| | Entropy | ResNet164 (Baseline) | 181 | $9.6 \pm 0.16$ | $6.4 \pm 0.27$ | $5.6 \pm 0.19$ | $218 \pm 1.4$ | $218 \pm 1.7$ | $218 \pm 1.6$ | $296 \pm 1.5$ | $338 \pm 2.2$ | $382 \pm 3.1$ |
| | | ResNet20 | 181 | $8.9 \pm 0.18$ | $5.7 \pm 0.23$ | $5.3 \pm 0.09$ | $24 \pm 1.3$ | $24 \pm 1.5$ | $25 \pm 1.5$ | $103 \pm 2.2$ | $145 \pm 1.3$ | $190 \pm 3.7$ |
| | | | 100 | $8.4 \pm 0.14$ | $5.6 \pm 0.17$ | $5.2 \pm 0.14$ | $14 \pm 1.1$ | $14 \pm 0.7$ | $14 \pm 0.7$ | $92 \pm 1.6$ | $134 \pm 1.2$ | $176 \pm 1.3$ |
| | | | 50 | $10.4 \pm 1.19$ | $6.3 \pm 0.55$ | $5.2 \pm 0.23$ | $7 \pm 0.8$ | $7 \pm 0.4$ | $7 \pm 0.4$ | $84 \pm 1.5$ | $126 \pm 1.6$ | $169 \pm 1.9$ |
| | | | 25 | $10.2 \pm 1.34$ | $6.2 \pm 0.31$ | $5.3 \pm 0.15$ | $4 \pm 0.3$ | $4 \pm 0.2$ | $4 \pm 0.2$ | $81 \pm 0.9$ | $123 \pm 1.2$ | $166 \pm 0.7$ |
| CIFAR100 | Forgetting Events | ResNet164 (Baseline) | 181 | $36.8 \pm 0.36$ | $27.1 \pm 0.40$ | $23.5 \pm 0.19$ | $221 \pm 6.1$ | $221 \pm 6.1$ | $221 \pm 6.1$ | $298 \pm 5.7$ | $342 \pm 5.5$ | $384 \pm 4.7$ |
| | | ResNet20 | 181 | $37.2 \pm 0.29$ | $27.1 \pm 0.14$ | $23.4 \pm 0.16$ | $24 \pm 0.7$ | $25 \pm 0.7$ | $25 \pm 0.7$ | $104 \pm 3.3$ | $148 \pm 3.6$ | $193 \pm 6.1$ |
| | | | 100 | $35.8 \pm 0.40$ | $27.7 \pm 0.24$ | $24.7 \pm 0.33$ | $14 \pm 0.3$ | $14 \pm 0.3$ | $14 \pm 0.3$ | $92 \pm 0.7$ | $134 \pm 0.6$ | $177 \pm 1.0$ |
| | | | 50 | $36.3 \pm 0.25$ | $28.2 \pm 0.24$ | $24.6 \pm 0.28$ | $8 \pm 0.4$ | $8 \pm 0.5$ | $8 \pm 0.5$ | $87 \pm 3.6$ | $132 \pm 6.2$ | $177 \pm 8.5$ |
| | | | 25 | $38.3 \pm 0.48$ | $28.4 \pm 0.32$ | $25.1 \pm 0.31$ | $4 \pm 0.2$ | $4 \pm 0.1$ | $4 \pm 0.1$ | $81 \pm 1.1$ | $123 \pm 1.2$ | $164 \pm 1.0$ |
| | Entropy | ResNet164 (Baseline) | 181 | $39.6 \pm 0.43$ | $30.1 \pm 0.12$ | $25.4 \pm 0.39$ | $220 \pm 6.4$ | $220 \pm 6.4$ | $220 \pm 6.4$ | $297 \pm 7.3$ | $340 \pm 7.3$ | $380 \pm 7.1$ |
| | | ResNet20 | 181 | $46.5 \pm 0.74$ | $29.7 \pm 0.45$ | $24.2 \pm 0.21$ | $24 \pm 0.6$ | $25 \pm 0.7$ | $25 \pm 0.7$ | $105 \pm 1.7$ | $148 \pm 2.6$ | $193 \pm 3.6$ |
| | | | 100 | $46.5 \pm 0.52$ | $29.7 \pm 0.36$ | $24.1 \pm 0.48$ | $14 \pm 0.4$ | $14 \pm 0.3$ | $14 \pm 0.3$ | $91 \pm 0.5$ | $135 \pm 0.7$ | $176 \pm 1.5$ |
| | | | 50 | $43.3 \pm 1.83$ | $30.0 \pm 0.77$ | $24.7 \pm 0.41$ | $7 \pm 0.2$ | $7 \pm 0.2$ | $8 \pm 0.2$ | $85 \pm 0.1$ | $128 \pm 1.3$ | $170 \pm 1.6$ |
| | | | 25 | $43.9 \pm 2.33$ | $30.8 \pm 1.37$ | $25.0 \pm 0.50$ | $4 \pm 0.1$ | $4 \pm 0.1$ | $4 \pm 0.1$ | $80 \pm 0.4$ | $123 \pm 0.9$ | $165 \pm 1.3$ |

A.6    ADDITIONAL CORRELATION RESULTS

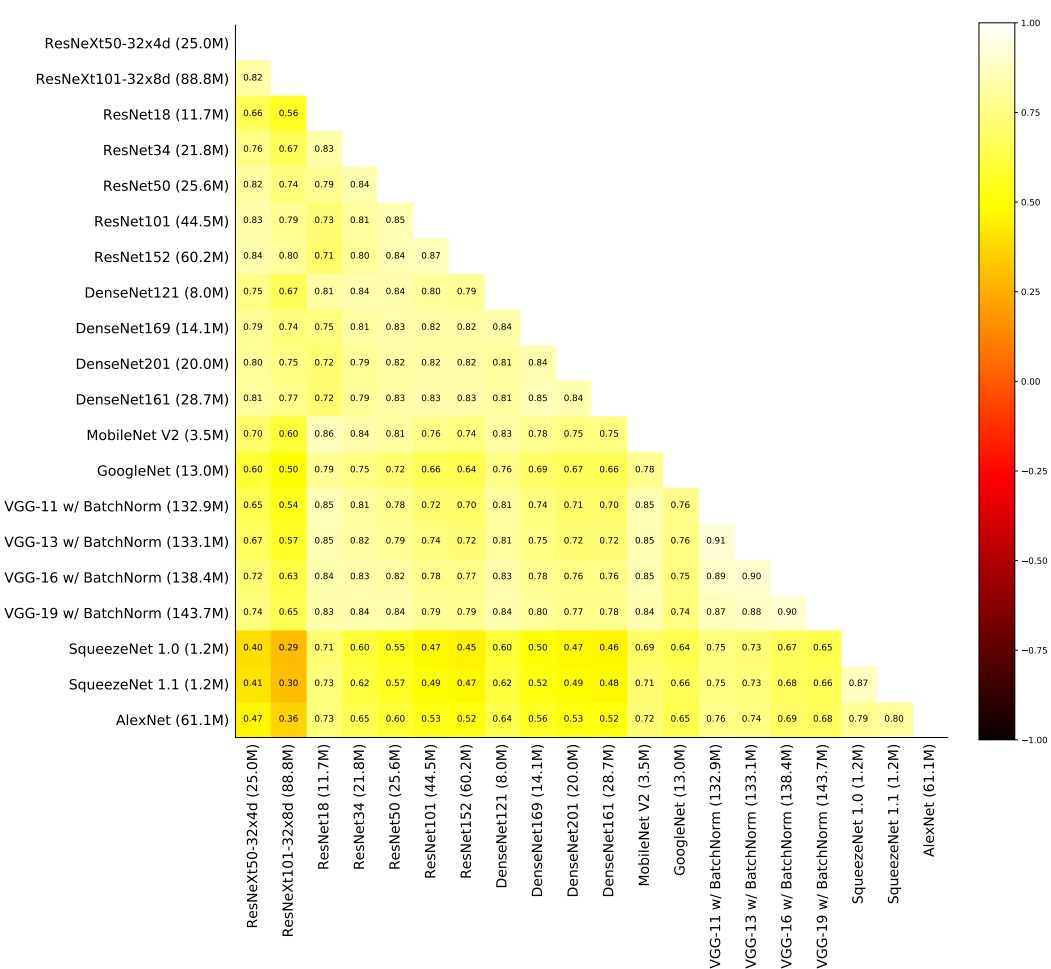

Figure 8: **Comparing selection across model architectures on ImageNet.** Spearman's correlation between max entropy rankings from PyTorch (Paszke et al., 2017) pretrained models on ImageNet. Correlations are high across a wide range of model architectures (Xie et al., 2017; He et al., 2016a; Sandler et al., 2018; Huang et al., 2017; Szegedy et al., 2015; Simonyan & Zisserman, 2014; Iandola et al., 2016; Krizhevsky et al., 2012). For example, MobileNet V2's entropy-based rankings were highly correlated to ResNet50, even though the model had far fewer parameters (3.5M vs. 25.6M). In concert with our fastText and VDCNN results from Section 3.2, the high correlations between different model architectures suggest that SVP might be widely applicable.

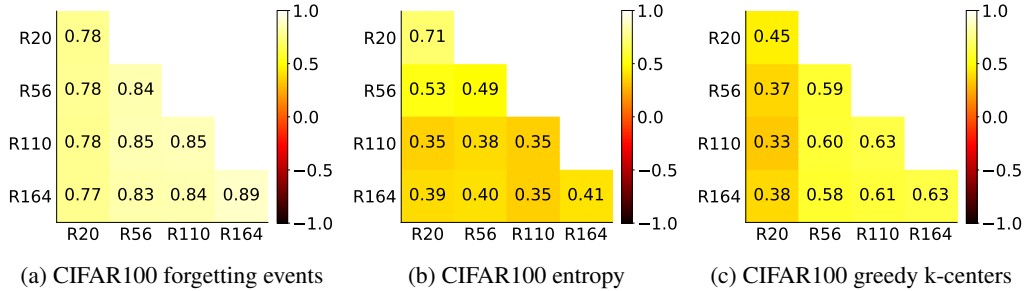

(a) CIFAR100 forgetting events      (b) CIFAR100 entropy      (c) CIFAR100 greedy k-centers

Figure 9: **Comparing selection across model sizes and methods on CIFAR100.** Average Spearman's correlation between different runs of ResNet (R) models and a varying depths. We computed rankings based on forgetting events (left), entropy (middle), and greedy k-centers (right). We saw a similarly high correlation across model architectures (off-diagonal) as between runs of the same architecture (on-diagonal), suggesting that small models are good proxies for data selection.

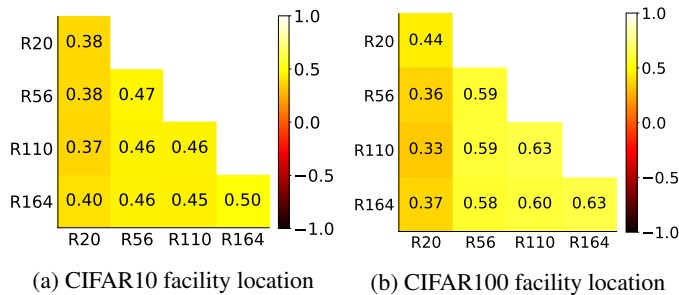

(a) CIFAR10 facility location      (b) CIFAR100 facility location

Figure 10: Spearman's rank-order correlation between different runs of ResNet (R) with pre-activation and a varying number of layers on CIFAR10 (left) and CIFAR100 (right). For each combination, we compute the average from 20 pairs of runs. For each run, we compute rankings based on the order examples are added in facility location using the same initial subset of 1,000 randomly selected examples. The results are consistent with Figure 3c and Figure 9c, demonstrating that most of the variation is due to stochasticity in training rather than the initial subset.

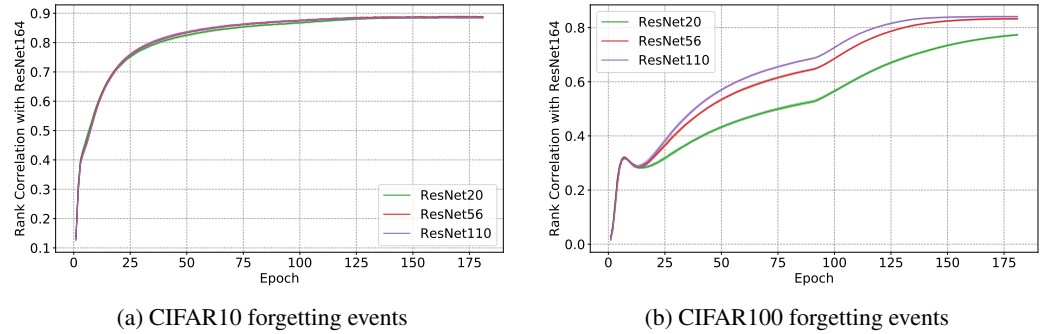

(a) CIFAR10 forgetting events      (b) CIFAR100 forgetting events

Figure 11: Average ($\pm$ 1 std.) Spearman's rank-order correlation with ResNet164 during 5 training runs of varying ResNet architectures on CIFAR10 (left) and CIFAR100 (right), where rankings were based on forgetting events.

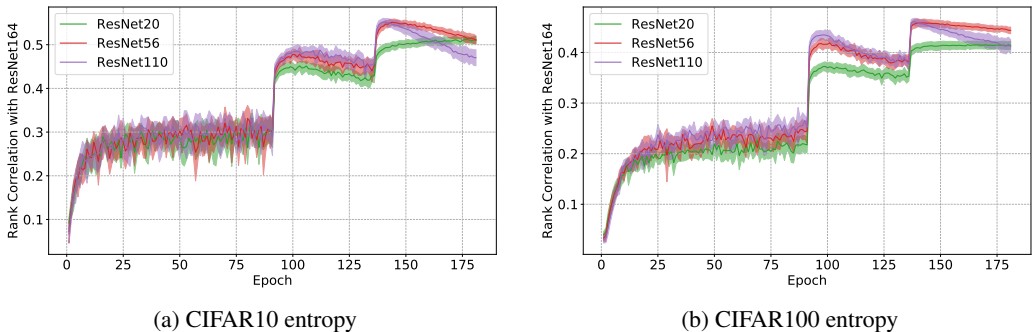

(a) CIFAR10 entropy

(b) CIFAR100 entropy

Figure 12: Average (± 1 std.) Spearman's rank-order correlation with ResNet164 during 5 training runs of varying ResNet architectures on CIFAR10 (left) and CIFAR100 (right), where rankings were based on entropy.

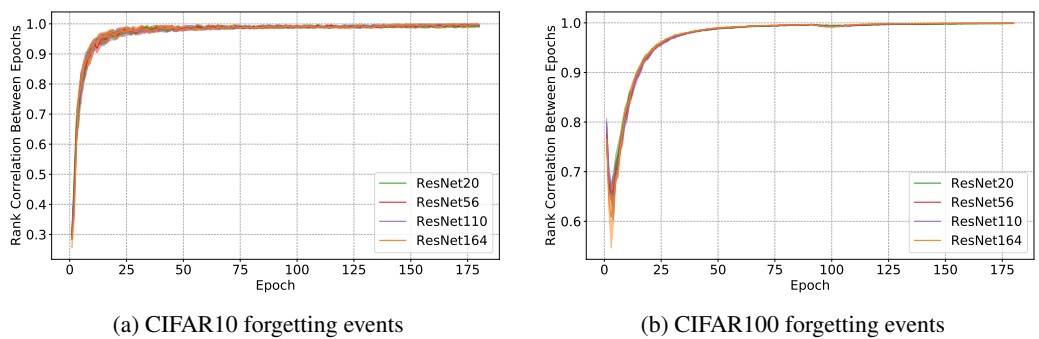

(a) CIFAR10 forgetting events

(b) CIFAR100 forgetting events

Figure 13: Average (± 1 std.) Spearman's rank-order correlation between epochs during 5 training runs of varying ResNet architectures on CIFAR10 (left) and CIFAR100 (right), where rankings were based on forgetting events.

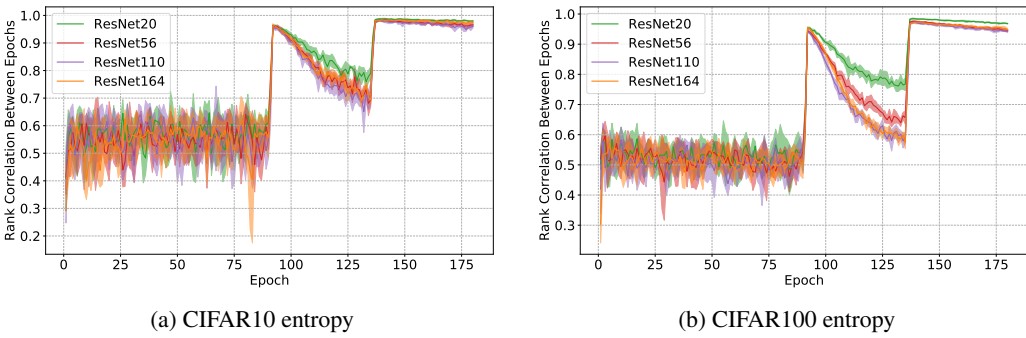

(a) CIFAR10 entropy

(b) CIFAR100 entropy

Figure 14: Average (± 1 std.) Spearman's rank-order correlation between epochs during 5 training runs of varying ResNet architectures on CIFAR10 (left) and CIFAR100 (right), where rankings were based on entropy.

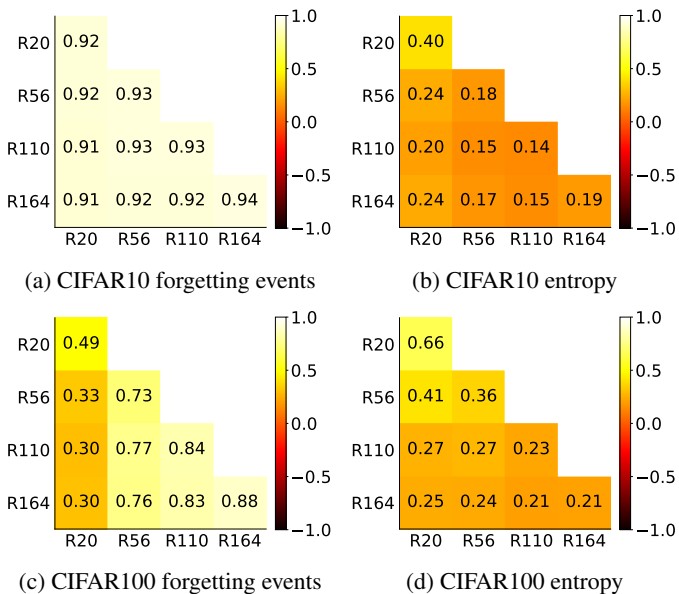

(a) CIFAR10 forgetting events      (b) CIFAR10 entropy

(c) CIFAR100 forgetting events      (d) CIFAR100 entropy

Figure 15: Pearson correlation coefficient between different runs of ResNet (R) with pre-activation and a varying number of layers on CIFAR10 (top) and CIFAR100 (bottom). For each combination, we compute the average from 20 pairs of runs. For each run, we compute rankings based on the number of forgetting events (left), and entropy of the final model (right). Generally, we see a similarly high correlation across model architectures (off-diagonal) as between runs of the same architecture (on-diagonal), providing further evidence that small models are good proxies for data selection.

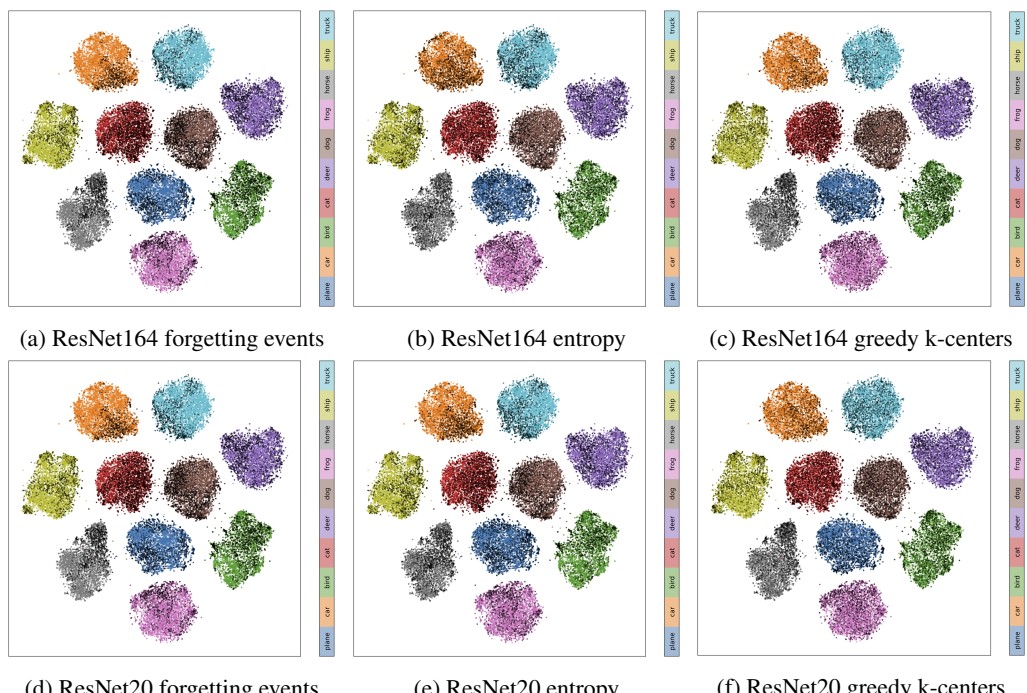

(a) ResNet164 forgetting events    (b) ResNet164 entropy    (c) ResNet164 greedy k-centers

(d) ResNet20 forgetting events    (e) ResNet20 entropy    (f) ResNet20 greedy k-centers

Figure 16: 2D t-SNE plots from the final hidden layer of a fully trained ResNet164 model on CIFAR10 and a 30% subset selected (black). The top row uses another run of ResNet164 to select the subset and the bottom row uses ResNet20. Rankings are computed using forgetting events (left), entropy (middle), and greedy k-centers (right).

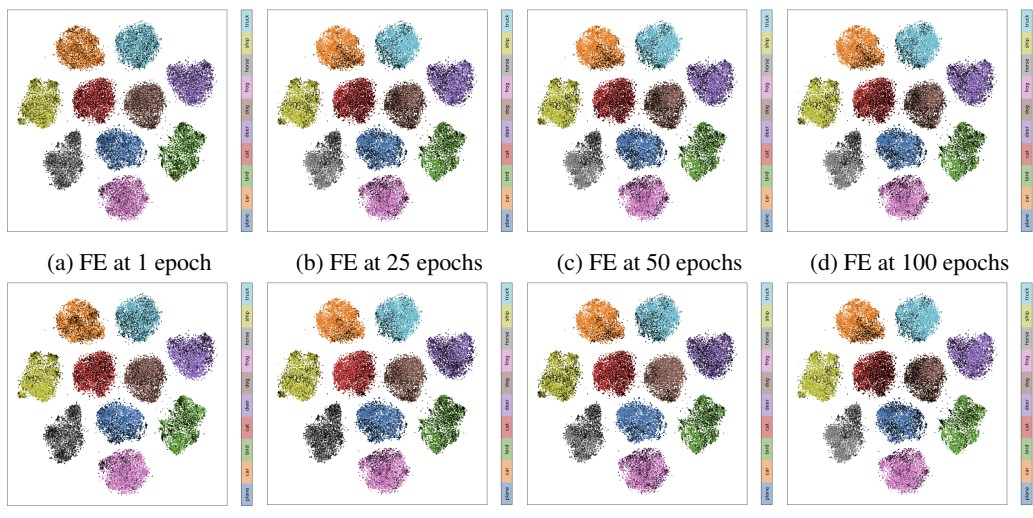

(a) FE at 1 epoch    (b) FE at 25 epochs    (c) FE at 50 epochs    (d) FE at 100 epochs

(e) Entropy at 1 epoch    (f) Entropy at 25 epochs    (g) Entropy at 50 epochs    (h) Entropy at 100 epochs

Figure 17: 2D t-SNE plots from the final hidden layer of a fully trained ResNet164 model on CIFAR10 and a 30% subset selected (black) with ResNet20 trained after a varying number of epochs. Rankings are calculated with forgetting events (top) and entropy (bottom). Notably, the ranking from forgetting events is much more stable because the model's uncertainty is effectively averaged throughout training rather than a single snapshot at the end like entropy. For t-SNE plots of the entire training run, please see http://bit.ly/svp-cifar10-tsne-entropy and http://bit.ly/svp-cifar10-tsne-forget for entropy and forgetting events respectively.

