# OpenReview forum: "Selection via Proxy: Efficient Data Selection for Deep Learning"
_ICLR.cc/2020/Conference — Accept (Poster)_

### Official Review · AnonReviewer2 · 2019-10-16
**Official Blind Review #2**

**Rating:** 6

**Review:**

This paper presents a method to speed up the data selection in active learning and core-set learning. The authors present a simple idea: instead of using the full model to select data points, they use a smaller model with fewer layers, potentially trained for fewer iterations. The authors show that this simple approach is able to speed up the data selection portion of both processes significantly with minimal loss in performance, and also results in significant speedup of the entire pipeline (data selection + training).

This paper is timely and important -- there has been a lot of emphasis lately on the environmental costs of training deep learning models (e.g., Strubell et al., ACL 2019; Schwartz et al., 2019 arxiv:1907.10597). This paper shows that simple, almost trivial techniques can lead to significant runtime benefits for active learning and core-set learning. The authors present an extensive set of experiments that validate their hypothesis, and the paper is overall clearly written.

Comments:

1. The correlation values in Figure 3 are quite diverse. It seemed "forgetting events" is much more correlated that the other two approaches.

2. Why do the authors think that SVP and their baselines failed to outperform random sampling on Amazon Review Full (towards the end of 3.3)?

3. Table 1 is very hard to interpret. I would advise the authors to look for a more succinct way to present their main findings. While this might seem contradictory, Figure 2, which is more reader-friendly, is also hard to interpret without the runtime values (nobody said visualization is easy!).

4. One thing that's missing from this paper is training the smaller networks end-to-end. What would be the effect of using the proxy network as the main network as well? this is likely to lead to very significant runtime savings, and I wonder at what costs.

Minor:

1. Towards the end of 3.2: I disagree that significant speedups are "uninteresting" if they lead to small reductions in performance.

**Experience Assessment:**

I have published one or two papers in this area.

**Review Assessment: Checking Correctness Of Derivations And Theory:**

N/A

**Review Assessment: Checking Correctness Of Experiments:**

I assessed the sensibility of the experiments.

**Review Assessment: Thoroughness In Paper Reading:**

I read the paper thoroughly.

---

> ### Author Response · Authors · 2019-11-15
> **Detailed Response**
>
> # R2P1: “Why do the authors think that SVP and their baselines failed to outperform random sampling on Amazon Review Full (towards the end of 3.3)?”
>
> Active learning is not guaranteed to work and depends on the specific task and dataset. Generally, on tasks with higher error, active learning achieves lower data efficiency (see Mussmann & Liang, ICML 2018). Amazon Review Full has relatively high error and is a very noisy dataset, so it is not too surprising that the baselines did not work. If the baseline approaches do not work, augmenting them with SVP is unlikely to improve performance because the proxies are, by definition, lower accuracy and give a rough approximation of the target model’s predictions.
>
> # R2P2: “Table 1 is very hard to interpret. I would advise the authors to look for a more succinct way to present their main findings. While this might seem contradictory, Figure 2, which is more reader-friendly, is also hard to interpret without the runtime values (nobody said visualization is easy!).”
>
> To make Figure 2 more reader-friendly, we added the speed-up each model gives over the baseline model to the legend. For Table 1, we removed some extraneous information to make it a little more readable and added the full table to the Appendix (see Table 3). We are happy to change it back, and we will continue to think about other ways to visualize the main findings from Table 1 and are open to suggestions.
>
> # R2P3: “One thing that's missing from this paper is training the smaller networks end-to-end. What would be the effect of using the proxy network as the main network as well? this is likely to lead to very significant runtime savings, and I wonder at what costs.”
>
> Figure 6 in the Appendix shows the accuracy the proxies achieve (dotted lines) for active learning. In all cases, the target model outperforms the proxy model in terms of error.

---

> ### Author Response · Authors · 2019-11-15
> **High-level Response**
>
> Thank you for your time and thoughtful feedback! We updated several tables and figures to make the paper more readable. Table 1, in particular, has been dramatically simplified to showcase the key results. We also added additional tables and figures to show trade-offs between accuracy and computational complexity better. Overall, thanks to your suggestions, the paper more clearly states the end-to-end training time savings and in turn, energy savings that can be achieved with a simple change to data selection techniques.
>
> Below we have provided a detailed response to each of your questions.

---

### Official Review · AnonReviewer3 · 2019-10-23
**Official Blind Review #3**

**Rating:** 6

**Review:**

What is the paper about ?
*The paper proposes a simple method to speed up active learning/core-set selection in deep learning framework. The idea is that instead of using full scale deep model for data selection/uncertainty sampling, use a smaller/faster proxy model and this proxy model’s uncertainty estimates are good enough for choosing data points even though proxy model’s accuracy may not be good.
*The paper experiments with 5 datasets with 2 active learning and 3 core set selection strategies and shows promising speed-up with little to no regression in performance in most settings.


What I like about this paper ?
*The idea is simple and the paper is well written and easy to follow with ample references.
*Unlike a lot of active learning papers, this paper considers both image and text domains.
*Results in both Table 1 and Figure  2 have been reported after multiple runs 3 and 5 respectively. Authors also report standard deviation which is important because the results can vary a lot from run to run in active learning.
*Plenty of empirical results do make the paper attractive and this definitely makes for a nice contribution to the active learning community.


What needs improvement ?
*Novelty is lacking. While the authors may be the first ones to apply this idea in the context of deep learning, they themselves note that the idea of using a smaller proxy like naive bayes for larger models like decision trees is not new. Although I agree this should not be a criteria for rejection especially given the extensive experimentation performed in the paper and has not impacted my score.
*The choice of datasets is not justified in the paper and could have been better. I don’t see the point of showing that method works well for both Amazon Review Polarity and Amazon Review Full. Same can be said for CIFAR10 and CIFAR100. Would have appreciated a more diverse set of datasets. I would have been happy to see results on WMT given data selection is an open challenge in machine translation.
*Paper initially proposes two methods for proxy: one by scaling down the model and two by reducing the number of epochs. fastext falls under neither of those categories and the giant speed up claims made by the paper like 41.7x are only with fastext. It is also important to note that while the performance drop using the fastext or smaller version of VDCNN is not much as compare to VDCNN, the performance gain is also not a lot compared to Random. Due to this reason I am not convinced if I should use SVP if I want speedup or just randomly select points especially in text domain.
*Would have liked to see the comparison of #parameters in your different proxy models instead of just names like VDCNN29 and VDCNN9.
*A lot of experimental details are missing which would make it difficult to reproduce the results unless the code is released.
*Why fasttext worked as a proxy for VDCNN and AlexNet did not work with ResNet is unclear from the paper.


Questions for Authors ?
*Do you plan to release the full code that can reproduce the results in the paper ? Based on my experience, I have found that a lot of active learning results are hard to reproduce and therefore it is difficult to draw any conclusions based on them.
*How sensitive are the results to hyper-parameter tuning and small things like choice of pre-trained word embeddings etc ? How did you select hyper-parameters at each stage during active learning ?


**Experience Assessment:**

I have published one or two papers in this area.

**Review Assessment: Checking Correctness Of Derivations And Theory:**

I assessed the sensibility of the derivations and theory.

**Review Assessment: Checking Correctness Of Experiments:**

I assessed the sensibility of the experiments.

**Review Assessment: Thoroughness In Paper Reading:**

I read the paper thoroughly.

---

> ### Author Response · Authors · 2019-11-15
> **Detailed Response to "What needs improvement?"**
>
> # R3P1: “The choice of datasets is not justified in the paper and could have been better.”
>
> We agree that we could have explained our choice of datasets better. To address this, we added additional details in the Appendix (A.1) that we reference in Section 3.1.
>
> Briefly, a crucial part of our rationale was that on tasks with lower error, active learning achieves higher data efficiency (see Mussmann & Liang, ICML 2018). We included both CIFAR10 (low-error) and CIFAR100 (high-error) to demonstrate that our approach performs as well as standard active learning at different points on the error and data efficiency curve. The Amazon Review datasets add a medium (text) and a much larger scale (3.6M and 3M examples, respectively). Adding ImageNet further allows us to investigate scale in the number of examples, but also the number of classes and the dimension of the input.
>
> # R3P2: “Would have appreciated a more diverse set of datasets. I would have been happy to see results on WMT given data selection is an open challenge in machine translation.”
>
> We agree that active learning for machine translation is an interesting and open problem, but we believe it is outside the scope of this paper. Active learning for machine translation is not as well-studied as classification, and the majority of the limited work on active learning for machine translation focuses on classical ML methods (e.g., statistical machine translation). In this paper, we intentionally focus on deep learning.
>
> Contemporary work on active learning for deep learning has primarily focused on image classification, and it is non-trivial to adapt these active learning methods from classification to machine translation. For example, many uncertainty sampling metrics (e.g., entropy, least confidence, and max-margin) assume a single categorical probability distribution for each example. However, machine translation is a sequential prediction task, where each example requires several separate predictions. While there are a small number of bespoke solutions for machine translation (Peris and Casacuberta, ACL 2018), we are focused on comparing against the broader body of active learning research.
>
> # R3P3: “Paper initially proposes two methods for proxy: one by scaling down the model and two by reducing the number of epochs. fastext falls under neither of those categories and the giant speed up claims made by the paper like 41.7x are only with fastext.”
>
> That is a great point. We made it more explicit in Section 2.3 by expanding the point about scaling down models to say that smaller architectures can also be considered. The rest of the text (e.g., abstract and introduction) was updated accordingly.
>
> # R3P4: “It is also important to note that while the performance drop using the fastext or smaller version of VDCNN is not much as compare to VDCNN, the performance gain is also not a lot compared to Random. Due to this reason I am not convinced if I should use SVP if I want speedup or just randomly select points especially in text domain.”
>
> Whether you should use SVP, and active learning more broadly, depends on the application. Ultimately, our SVP approach shows another point on the trade-off curve between speed (random sampling wins) and accuracy (our method wins). At 20% on Amazon Review Polarity, SVP with VDCNN9 achieves about 0.7% lower error (i.e., a 12.5% relative error reduction) compared to random sampling. This is a 2x improvement in data efficiency over random sampling, which did not reach the same error until 40%. With fastText, SVP likewise significantly improves data efficiency. While a 12.5% relative error reduction might seem minor, one could imagine critical text classification problems where this could be a worthwhile trade-off.
>
> # R3P5: “Would have liked to see the comparison of #parameters in your different proxy models instead of just names like VDCNN29 and VDCNN9.”
>
> Great point! We added Table 2 to the appendix with the number of parameters for the main models in the paper. For the figure comparing selection across model architectures on ImageNet (now Figure 8), we included the number of parameters for each pretrained model as part of the label.
>
> # R3P6: “Why fasttext worked as a proxy for VDCNN and AlexNet did not work with ResNet is unclear from the paper.”
>
> We investigated this further and found an error in our script that generated the rankings from PyTorch’s pretrained model. We updated Figure 8 with the correct correlations and changed Section 3.4 to reflect the new results. AlexNet and ResNet are much more correlated than we originally thought, so it might be a viable proxy. However, MobileNet V2 has an even higher correlation to ResNet50 (on par with ResNet18) and far fewer parameters (3.5M). While we did not have time to run this experiment for the rebuttal, the high correlations suggest that SVP might be even more robust than we initially presented.

---

> > ### Author Response · Authors · 2019-11-15
> > **Detailed Response to "Questions for Authors?"**
> >
> > # R3P7: “Do you plan to release the full code that can reproduce the results in the paper ? Based on my experience, I have found that a lot of active learning results are hard to reproduce and therefore it is difficult to draw any conclusions based on them.”
> >
> > Great point! We take reproducibility very seriously and have added more details to Section 3.2 to clarify details like the initial random subset and selection size. We will also release the code on Github if the paper is accepted. In the meantime, we have included the working code in a separate private comment.
> >
> > # R3P8: “How sensitive are the results to hyper-parameter tuning and small things like choice of pre-trained word embeddings etc ? How did you select hyper-parameters at each stage during active learning ?”
> >
> > To avoid extensive hyperparameter tuning, we mostly followed the same training procedure (e.g., optimizer, learning rate, momentum, data augmentation, and batch size) as the papers for each model we considered, as described in section A.2. Empirically, the learning rate schedule for these models have been robust to training variants (e.g., model architectures, floating-point precisions, and batch sizes).
> >
> > With respect to pre-training, neither VDCNN nor fastText use pre-trained word embeddings. VDCNN is a character-level model, and fastText learns its embeddings from scratch.

---

> ### Author Response · Authors · 2019-11-15
> **High-level Response**
>
> Thank you for your time and thoughtful feedback! We clarified many details in the paper and packaged the code to share. During our investigation, we found an error in the script that generated the rankings from PyTorch’s pretrained model. Rankings were much more highly correlated across a wide range of models (Figure 8 in the Appendix). For example, MobileNet V2’s entropy-based rankings were highly correlated to ResNet50 (on par with ResNet18), even though the model had far fewer parameters (3.5M vs. 25.6M). In concert with our fastText and VDCNN results, the high correlations between different model architectures suggest that SVP might be even more robust than we initially presented. Your suggestions were the catalyst for these findings!
>
> Below we have provided a detailed response to each of your questions.

---

> ### Public Comment · ~Jason_Xiaotian_Dou2 · 2023-02-06
> **idea of using a smaller proxy like naive bayes for larger models like decision trees**
>
> Dear reviewer,
>
> Could you please give a pointer on what paper this is: "using a smaller proxy like naive bayes for larger models like decision trees"?

---

### Official Review · AnonReviewer4 · 2019-11-07
**Official Blind Review #4**

**Rating:** 6

**Review:**

The paper proposes a method for selecting a subset of a large dataset to reduce the computational costs of deep neural netwoks. The main idea is to train a proxy model, a smaller version of the full neural network, to choose important data points for active learning or core-set selection. Experiments on standard classification tasks demonstrate that this approach can yield substantial computational savings with only a small drop in accuracy.

This paper is well-written and was easy to follow, with a clear motivation. The paper does good job of demonstrating that the proposed algorithm is effective through a comprehensive set of experiments. Overall, I favor acceptance and would be willing to increase my score if the following are addressed:

1) Training for a smaller number of epochs was mentioned as a possibility to save computation. In the experiments, this was only done for one of the settings. Is this because models trained for fewer epochs are ineffective for data selection?
Looking at Fig. 5b, it seems like the error drops significantly around 14 min before plateauing. In practice, for a new dataset, it could be difficult (or impossible) to know when to stop training a proxy a priori so that it achieves good performance relative to a larger model. It could be interesting to look at the effectiveness of a proxy at various points during training to see if the benefits are sensitive to the training time.

2) For the active learning experiments, how many data points were added to the training set in each round? I could not find this number in the paper. Also, how many rounds of data selection were there?

3) Is there an explanation why in Fig. 2b and Fig. 7c, the resnet20 proxy performs better than the larger (and more accurate) models? In particular, it even outperforms the 'oracle' baseline, which I find surprising.

4) On a similar note, in Fig. 7, for small subsets (30%), the random baseline outperforms all of the SVP settings. Why is SVP ineffective in this case?

Minor comments/suggestions:
- In the abstract: "improvement in data selection runtime" Is this "data selection runtime" different from the total runtime? If not, it could be clearer to simply state it as the "total runtime (including the time to repeatedly train and select points)". I was unsure if this was a different measure.

- Did the authors try further reducing the model capacity? With the success of the smaller models, it seems natural to try pushing further in this direction.

- Since one of the findings was that models with similar architectures were effective as proxies but not models with different ones, perhaps there could be even higher correlation if the proxies were initialized with the exact same weights as a subset of the full model.

- The writing in Table 1 (and the ones in the appendix) is a bit small.



**Experience Assessment:**

I do not know much about this area.

**Review Assessment: Checking Correctness Of Derivations And Theory:**

N/A

**Review Assessment: Checking Correctness Of Experiments:**

I assessed the sensibility of the experiments.

**Review Assessment: Thoroughness In Paper Reading:**

I read the paper thoroughly.

---

> ### Author Response · Authors · 2019-11-15
> **Detailed Response on Major Points**
>
> # R4P1: “Training for a smaller number of epochs was mentioned as a possibility to save computation. In the experiments, this was only done for one of the settings. Is this because models trained for fewer epochs are ineffective for data selection?"
>
> Great point and question! To explore the effectiveness of training for fewer epochs in other settings, we ran additional active learning experiments on CIFAR10 and CIFAR100. The results show that training for fewer epochs can provide a significant improvement over random sampling but is not quite as good as training for the full 181 epochs, as shown in Table 4 in the Appendix. Additionally, we added training for 25 epochs to our core-set experiments in Table 7 in the Appendix. While we were unable to finish experiments on the more computational intensive datasets before the rebuttal deadline, we will add more results in the next version of the paper to provide more evidence about training for fewer epochs.
>
> In general, the effectiveness of partial training depends on the stability of the selection metric. Figures 11 and 12 in the Appendix show how quickly proxy rankings converge to the target model. Forgetting events converges much faster, especially on CIFAR10, than entropy, which makes training for fewer epochs more effective. Figure 17 in the Appendix illustrates this by visualizing selected points in 2D t-SNE plots (and videos). While metrics like greedy k-centers and entropy use a single snapshot of the model, forgetting events effectively averages the model’s uncertainty over time to get a more stable and accurate ranking.
>
> Of course, we cannot compare the proxy’s ranking to the target model to determine when to stop in practice, but there are two ways to deal with this. For large-scale applications where data is continuously collected based on user interactions, models are often re-trained periodically on the most recent data, and the ideal stopping point can be determined from historical data. Secondly, the change in rank correlation between adjacent epochs provides a useful signal. Figures 13 and 14 in the Appendix show the average rank correlation between adjacent epochs for CIFAR10 and CIFAR100. For a new dataset, this could be calculated after every epoch like validation accuracy and used to tell when the proxy’s ranking has successfully stabilized. In practice, it is very similar to early stopping.
>
> # R4P2: “For the active learning experiments, how many data points were added to the training set in each round? I could not find this number in the paper. Also, how many rounds of data selection were there?”
>
> For active learning, we follow the same procedure as Sener & Savarese. Starting with an initial random subset of 2% of the data, we selected 8% of the remaining unlabeled data for the first round and 10% for subsequent rounds until the labeled data reached the budget b and retrained the models from scratch between rounds, as described in Section 2.1. To make this clear in the paper, we added an introductory paragraph to Section 3.2.
>
> # R4P3: “Is there an explanation why in Fig. 2b and Fig. 7c, the resnet20 proxy performs better than the larger (and more accurate) models? In particular, it even outperforms the 'oracle' baseline, which I find surprising.”
>
> We do not have a definitive answer on this, but one possible explanation could be that ResNet20’s predictions are more consistent throughout the course of training than the larger models. Looking at Figures 12 and 14, the entropy-based rankings on CIFAR10 change more for large models than small proxies like ResNet20. These large changes in entropy can cause points that are ambiguous to be mischaracterized as uninformative, leading to lower data efficiency and less correlation on average.
>
> Notably, forgetting events does not have this problem (Figures 11 and 13) because the model’s uncertainty is effectively averaged throughout training rather than a single snapshot at the end like entropy or greedy k-centers. To demonstrate, the videos in Figure 17 show how volatile model’s decision boundary can be during training.
>
> # R4P4: On a similar note, in Fig. 7, for small subsets (30%), the random baseline outperforms all of the SVP settings. Why is SVP ineffective in this case?
>
> The random baseline also outperforms the oracle baseline, so this is not a problem with SVP, but rather the underlying selection methods. The point of our paper is to show that we can accelerate existing methods by substituting in a computationally inexpensive proxy model for the more accurate target model during data selection. If the underlying selection method does not work, there is no guarantee that SVP will work.

---

> > ### Author Response · Authors · 2019-11-15
> > **Detailed Response on Minor Points**
> >
> > # R4P5: “- In the abstract: "improvement in data selection runtime" Is this "data selection runtime" different from the total runtime? If not, it could be clearer to simply state it as the "total runtime (including the time to repeatedly train and select points)". I was unsure if this was a different measure.”
> >
> > Data selection runtime is different from the total runtime. Data selection runtime is the time it takes to repeatedly train and select points and does not include the time to train the final target model. This includes everything in the dotted line boxes in Figure 1. For core-set selection, we include both data selection runtime and total runtime in Tables 6 and 7. For active learning, we only include data selection runtime because this becomes the dominant component for larger budgets or smaller selection sizes.
> >
> > # R4P6: “Did the authors try further reducing the model capacity? With the success of the smaller models, it seems natural to try pushing further in this direction.”
> >
> > There may be some cases where even smaller models may serve as reasonable proxies. However, in many of our experiments, we found little or no computational performance gains from using smaller proxy models. On ImageNet, for example, ResNet18 was already small enough that disk latency was the bottleneck rather than compute, at least on our hardware. Any smaller model would not have reduced runtime. On the Amazon Review datasets, fastText was already extremely efficient, taking less than 10 minutes to train on a CPU. Any improvements to the proxy on Amazon Review would result in a relatively minimal improvement in overall training time, far overshadowed by the much longer time it takes to train a large model such as VDCNN29 on the examples selected by fastText. For CIFAR10 and CIFAR100, smaller ResNet models did not run faster, likely due to data transfer and kernel launch overhead. Further reducing the model size would increase the proxy’s error without a significant speed-up. To illustrate, we added smaller ResNets to Figures 4a and 5a.
> >
> > # R4P7: “Since one of the findings was that models with similar architectures were effective as proxies but not models with different ones, perhaps there could be even higher correlation if the proxies were initialized with the exact same weights as a subset of the full model.”
> >
> > This comment was a little unclear to us. If you meant just transfering weights for the correlation plots, we tried to limit the influence weight initialization would have on correlation by only using *different* runs/initializations for the baseline correlation of each model. Rather than just reporting 1.0 for the baseline correlation, this approach includes the variance from random initializations and data ordering, giving a fair baseline for comparing correlations across model architectures.
> >
> > Alternatively, maybe you meant transfer weights from the proxy to the full model. Initializing the full model with weights from the proxy is a possibility, but there is not a standard way to partially warm start models like this. Moreover, we wanted to enable a fair comparison and isolate the impact of SVP, so we followed the same procedure as prior work, which trained from scratch, as mentioned in Section 2.1. Thoroughly investigating warm starting and partial initializations would be an exciting direction for future work.
> >
> > # R4P8: “The writing in Table 1 (and the ones in the appendix) is a bit small.”
> >
> > For Table 1, we removed some extraneous information to make it a little more readable and added the full table to the Appendix (see Table 3). We are happy to change it back if you prefer the original version. For Tables 2 and 3 (now Tables 6 and 7) in the Appendix, we displayed them in landscape. Table 4 (now Table 5) had fewer columns than Tables 2 and 3 (now Tables 6 and 7), so the text was larger, and we kept that as is.

---

> ### Author Response · Authors · 2019-11-15
> **High-level Response**
>
> Thank you for your time and thoughtful feedback!  We ran additional experiments to explore the effectiveness of training for fewer epochs in a variety of settings. We also created more plots to explain how the rankings from proxies compared to the larger target model. These results show that the underlying selection method plays an essential role in the effectiveness of SVP. Notably, the rankings from forgetting events are much more stable because the model’s uncertainty is effectively averaged throughout training rather than a single snapshot at the end, like entropy or greedy k-centers. Based on your questions and the points you raised, the paper more clearly presents the underlying dynamics that affect performance.
>
> Below we have provided a detailed response to each of your questions.

---

### Decision · Program_Chairs · 2019-12-19

**Decision:**

Accept (Poster)

**Comment:**

This paper proposes to perform sample selection for deep learning - which can be very computationally expensive - using a smaller and simpler proxy network. The paper shows that such proxies are faster to train and do not substantially harm the accuracy of the final network.

The reviewers were all in agreement that the problem is important, and that the paper is comprehensive and well executed. I therefore recommend it should be accepted.